# Intrafusal-fiber LRP4 for muscle spindle formation and maintenance in adult and aged animals

Rangjuan Cao [1,2,4], Peng Chen[1,4], Hongsheng Wang [1,4], Hongyang Jing[1], Hongsheng Zhang [1], Guanglin Xing[1], Bin Luo [1], Jinxiu Pan[1], Zheng Yu[1], Wen-Cheng Xiong[1,3] ✉ & Lin Mei [1,3] ✉

Proprioception is sensed by muscle spindles for precise locomotion and body posture. Unlike the neuromuscular junction (NMJ) for muscle contraction which has been well studied, mechanisms of spindle formation are not well understood. Here we show that sensory nerve terminals are disrupted by the mutation of *Lrp4*, a gene required for NMJ formation; inducible knockout of *Lrp4* in adult mice impairs sensory synapses and movement coordination, suggesting that LRP4 is required for spindle formation and maintenance. LRP4 is critical to the expression of Egr3 during development; in adult mice, it interacts in trans with APP and APLP2 on sensory terminals. Finally, spindle sensory endings and function are impaired in aged mice, deficits that could be diminished by LRP4 expression. These observations uncovered LRP4 as an unexpected regulator of muscle spindle formation and maintenance in adult and aged animals and shed light on potential pathological mechanisms of abnormal muscle proprioception.

The movement and position of muscles are controlled by several synapses in the muscle. Muscle contraction is mediated by the neuromuscular junction (NMJ) between α-motor neurons (α-MN) and skeletal muscle fibers (referred to as α-MN NMJs)[1–3]. The precise locomotion and limb and axial position are sensed by mechanoreceptors such as spindles that are embedded in muscles[4,5]. Muscle spindles consist of intrafusal fibers that are innervated by afferent axons of DRG neurons that coil around their equatorial regions and form sensory synapses with characteristic "annulospiral endings"[6–8]. They relay muscle stretch signals to the brain for proprioception which is critical to the coordination of movement, posture and body awareness[4,5]. Intrafusal fibers are also innervated by γ-motor neuron (γ-MN) axons to form γ-MN NMJs, which via activating AChRs control tension of intrafusal fibers and thus maintain the stretch sensitivity of spindles[9–11] (Fig. 1a). Malfunction of muscle spindles leads to uncoordinated gaits, abnormal limb positioning and resting tremors and has been implicated in neurological disorders including muscular dystrophy, movement disorders such as

Parkinson's disease, Huntington's disease, multiple sclerosis, diabetic peripheral neuropathy as well as ageing[8,12,13].

The α-MN NMJ formation requires agrin, a factor from α-motor axons, which binds the extracellular domain (ECD) of LRP4, a member of the low-density lipoprotein receptor (LDLR) family[14], on muscle fibers to activate the transmembrane tyrosine kinase MuSK[15–17]. Mutation of *agrin*, *Lrp4* or *MuSK* causes neonatal lethality because of NMJ defects[18–20]. The agrin-LRP4-MuSK signaling is also critical for α-MN NMJ maintenance[21–23]; its gain-of-function improves α-MN NMJ function in mouse models of neuromuscular disorders, ageing, and regeneration[24–27]. Mutation of *agrin* or *MuSK* disrupts AChR aggregation on intrafusal fibers[28]. These results indicate that the agrin-LRP4-MuSK signaling is necessary for the formation of both α-MN and γ-MN NMJs.

Muscle spindle formation begins at embryonic day (E) 14.5-E15.5 in mice when several slow muscle fibers become innervated by Ia afferent axons; subsequently, these fibers differentiate into intrafusal muscle fibers. Around birth, intrafusal fibers are innervated by γ-motor axons to

[1]Department of Neurosciences, School of Medicine, Case Western Reserve University, Cleveland, OH 44106, USA. [2]Department of Hand and Foot Surgery, China-Japan Union Hospital of Jilin University, Changchun, China. [3]Louis Stokes Cleveland Veterans Affairs Medical Center, Cleveland, OH 44106, USA. [4]These authors contributed equally: Rangjuan Cao, Peng Chen, Hongsheng Wang. ✉e-mail: wxx119@case.edu; lin.mei@case.edu

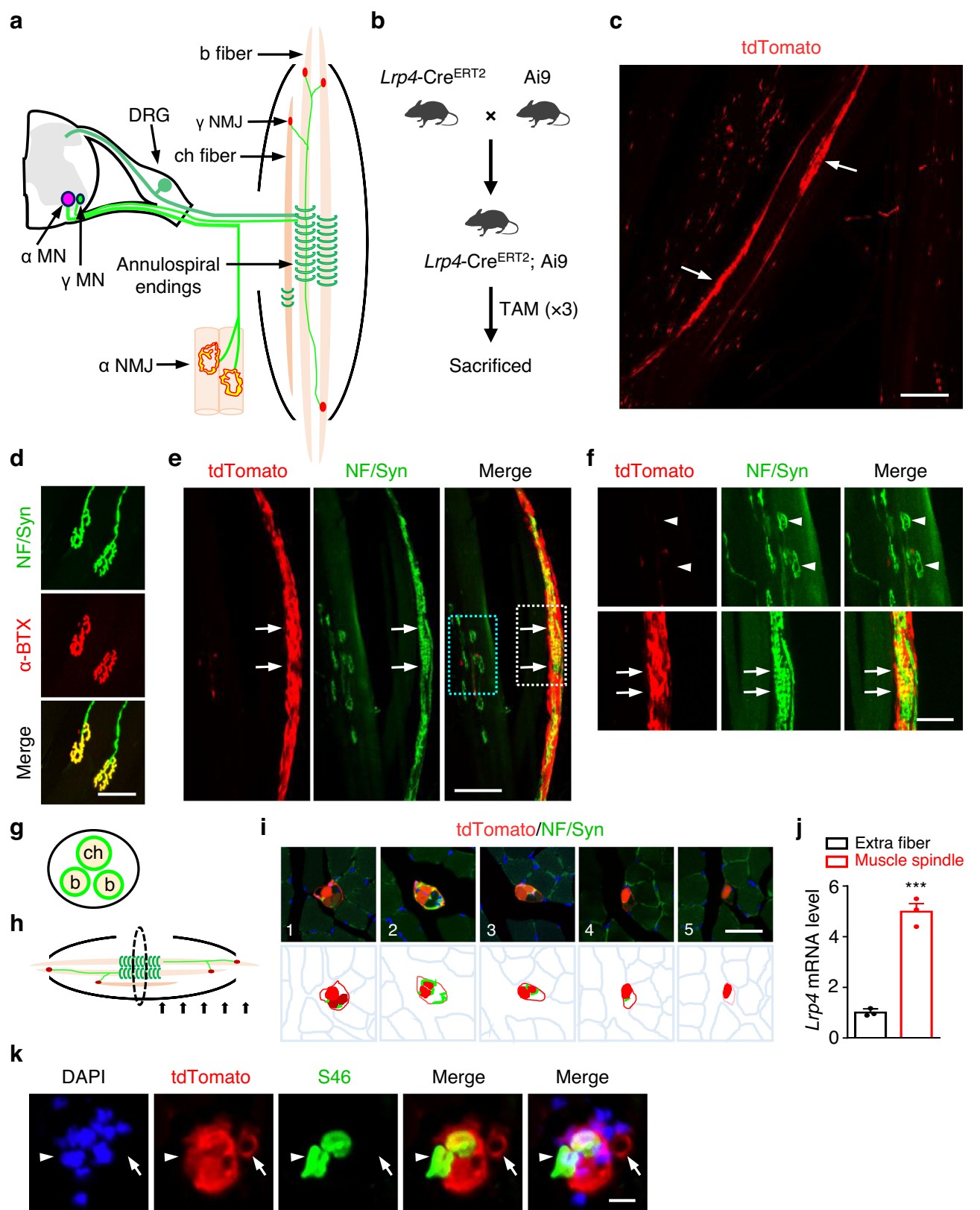

form γ-MN NMJs[29]; muscle spindles continue to develop after birth. Unlike α-MN NMJs that have been studied extensively, less is known regarding the development and maintenance of muscle spindles. Interestingly, *agrin* or *MuSK* mutation has little effect on the morphology of annulospiral sensory endings at E18.5, suggesting a dispensable role of the classic agrin signaling in embryonic sensory synapse formation[28]. However, spindle formation requires early growth

response 3 (Egr3), a zinc-finger transcription factor that is transiently expressed in intrafusal fibers (beginning ~E15.5, but not in adulthood)[30,31]; *Egr3* mutant mice are impaired in intrafusal muscle differentiation and spindle development or function[30]. Egr3 expression is induced by several factors released by Ia afferent axon terminals, including Ig-neuregulin 1 (Ig-Nrg1). Via activating ErbB2/3 receptors on slow muscle fibers, Ig-Nrg1 promotes intrafusal fiber differentiation and

**Fig. 1 | High expression of LRP4 in intrafusal fibers. a** Schematic diagram of muscle spindle and NMJ. γ MN, γ motor neuron; α MN, α motor neuron; DRG, dorsal root ganglia; b fiber, bag fiber; ch fiber, chain fiber. **b** *Lrp4*-Cre[ERT2] crossed with reporter mice Ai9 and the schedule of tamoxifen administration. **c** tdTomato labeled a few fibers of leg muscles in *Lrp4*-Cre[ERT2]; Ai9 mice. Scale bar, 200 μm. This experiment was repeated three times independently with similar results. **d** Anti-NF/ Syn antibodies labeled pretzel-like structures positive for CF568-labeled α-BTX. Scale bar, 50 μm. This experiment was repeated three times independently with similar results. **e–i** The enrichment of tdTomato in muscle spindle. **e** Representative images of muscle spindle in leg muscle of *Lrp4*-Cre[ERT2]; Ai9 mice at P14 stained by NF/ Syn. Scale bar, 100 μm. **f** Enlarged images of blue rectangle and white rectangle in (**e**). Arrows, annulospiral endings; arrowheads, regular NMJs. Scale bar, 50 μm. This experiment was repeated three times independently with similar results. **g** The

schematic diagram of transection view of a muscle spindle. **h** The schematic diagram of series transection view in (**i**). **i** Tdtomato was concentrated in the intrafusal fibers and capsules. Images of transection view of a muscle spindle in *Lrp4*-Cre[ERT2]; Ai9 leg muscle (P60) at different locations. The internal distance is 100 μm. Scale bar, 40 μm. This experiment was repeated three times independently with similar results. **j** Elevated expression of *Lrp4* mRNA in muscle spindle. Tdtomato[+] muscle spindles were isolated and proceed to quantitative RT-PCR analysis. The related *Lrp4* mRNA was compared with the extrafusal fibers ($t_{(4)}$ = 12.046, ***$p$ = 0.0003). Data are shown as mean ± SEM. $n$ = 3 mice per group, unpaired two-tailed $t$ test. **k** Representative images of transection view of *Lrp4*-Cre[ERT2]; Ai9 leg muscle (P60) stained by DAPI and S46. Scale bar, 10 μm. This experiment was repeated three times independently with similar results. Source data are provided as a Source Data file.

thus spindle formation[32–34]. Mutation of *Bace1* that is required for releasing Nrg1 results in abnormal muscle spindle[35]. On the other hand, intrafusal muscle fibers release NT-3 to promote the survival of sensory neurons by activating TrkC; mice lacking either of NT-3 or TrkC fail to form functional muscle spindles[36,37].

In this study, we show that LRP4 is expressed in intrafusal muscle fibers. Sensory nerve terminals were disorganized, discontinuous or fragmented in *Lrp4 null* mutant mice, demonstrating a role in spindle development. Inducible knockout of *Lrp4* in adult mice caused the loss of annulospinal endings, disrupted sensory synapses, and compromised movement coordination, indicating that LRP4 is required for the maintenance of muscle spindles. We show that spindle sensory endings and function are impaired in aged mice; these deficits could be diminished by LRP4 expression. Finally, because agrin and MuSK are not involved in spindle development, we investigated how LRP4 regulates spindle development and maintenance. Our results suggest that LRP4 of intrafusal fibers is critical to the expression of Egr3 during development; in adult mice, it interacts in trans with amyloid precursor protein (APP) and amyloid precursor like protein 2 (APLP2) of sensory terminals. Together, these observations demonstrate a role of LRP4 in muscle spindle formation and maintenance in adult and aged animals. By identifying a previously unrecognized function of LRP4, the results shed light on potential pathological mechanisms of diseases with abnormal muscle proprioception.

## Results

### LRP4 expression in intrafusal fibers

To study LRP4 expression in muscles, we crossed *Lrp4*-Cre[ERT2] mice where Cre[ERT2] was driven by the endogenous *Lrp4* promoter[38], with the reporter line Ai9. Resulting compound mice (*Lrp4*-Cre[ERT2]; Ai9) were injected with tamoxifen (TAM) (Fig. 1b) to induce the expression of tdTomato. We showed previously tdTomato specifically label cells where LRP4 promoter is active including astrocytes in the brain[38]. *Lrp4*-Cre[ERT2]; Ai9 mice are able to form the NMJs and displayed normal survival rates. Unexpectedly, the tdTomato fluorescence signal labeled a few muscle fibers with high intensity (Fig. 1c). To determine the identity of tdTomato[+] fibers, limb muscles were stained whole-mount with antibodies against neurofilament (NF) and synapsin (Syn) to label axons and nerve terminals. Staining with anti-NF/Syn antibodies revealed pretzel-like structures that were stained with CF568-α-BTX (α-BTX in short) that labels the AChR (Fig. 1d), suggesting that these structures are α-MN NMJs and validating the specificity of the antibodies. Intriguingly, anti-NF/Syn antibodies also labeled tdTomato[+] fibers (Fig. 1e); in particular, in the middle of these fibers, nerve terminals displayed annulospiral endings, characteristic of muscle spindles (Fig. 1a, e, f), suggesting that LRP4 is highly expressed in intrafusal fibers (Fig. 1e, f, arrows). To further test this hypothesis, series cross-sections of intrafusal fibers were prepared beginning from the nerve terminal-rich (or equatorial) region to the tendon (or polar) region (Fig. 1g, h). As shown in Fig. 1i, tdTomato fluorescence was detected in intrafusal fibers and capsule cells. To further determine the expression of LRP4 in intrafusal fibers,

tdTomato[+] muscle spindles were isolated and subjected to quantitative RT-PCR. *Lrp4* mRNA was about fivefold higher in muscle spindles, compared with extrafusal fibers (Fig. 1j). However, the intensity of tdTomato fluorescence appeared to be variable among bag (i.e., S46[+]) fibers and among S46[-] fibers (presumably chain intrafusal fibers) (Fig. 1k). tdTomato was not visible in extrafusal fibers (Fig. 1f, arrowheads), presumably because the LRP4 promoter is active only in a handful of synaptic nuclei of multi-nucleated extrafusal fibers[39]. Together, these results suggest that LRP4 is expressed in intrafusal fibers.

### LRP4 in spindle development

The finding of LRP4 in intrafusal fibers prompted us to investigate whether LRP4 regulates spindle development. To this end, we studied *Lrp4* mutant mice, whose *Lrp4* gene was replaced by the *LacZ* gene[40]. Homozygous *LacZ/LacZ* mice (referred to as *Lrp4*[−/−] hereafter) failed to form α-MN NMJs and died neonatally[39,40] (Supplementary Fig. 1a). Therefore, we characterized mice at E18.5 when sensory nerve terminals began to develop and have not formed annulospiral endings[29,41,42]. Muscles were whole-mount stained with an antibody against vesicular glutamate transporter 1 (VGluT1) that labels sensory nerve terminals[43] and with NF/Syn antibodies. Anti-VGluT1 antibody labeled a central segment that co-stained with NF/Syn antibodies, indicative of spindle nerve terminals (Fig. 2a, b). Remarkably, the length of VGluT1[+] segments was reduced in *Lrp4*[−/−] mice, compared with control mice (Fig. 2c). Moreover, some NF/Syn[+] terminals extended into regions that were not stained with VGluT1 (Fig. 2b, arrows). These results demonstrate that the sensory endings are impaired in the absence of LRP4. Interestingly, unlike control mice where NF/Syn[+] nerve terminals were well organized, those in *Lrp4* mutant mice were thinner, shorter and poorly organized (Fig. 2a, arrows). Besides, NF/Syn[+] axon terminals appeared fragmented (Fig. 2a, arrowheads) and shorter in mutant mice (Fig. 2d). These results suggest that the development of axons and nerve terminals is compromised in *Lrp4*[−/−] mice. To determine whether *Lrp4* mutation altered intrafusal muscle development, muscle cross-sections were stained for NF/Syn and S46. As shown in Fig. 2e–g, the number and size of S46[+] bag fibers were comparable between *Lrp4*[+/−] and *Lrp4*[−/−] mice. These results suggest that *Lrp4* mutation impairs development of sensory endings.

### Reduced Egr3 levels in *Lrp4* mutant mice

Next, we investigated the impact of *Lrp4* mutation on sensory nerve innervation by staining for parvalbumin (PV), a marker of sensory axons and terminals[31,37]. In addition, we also characterized the expression of Egr3 (Fig. 3a–c). Muscle spindles form between E14.5-15.5 when sensory nerves begin to innervate muscle fibers[36,44]. In agreement, as shown in Fig. 3a, PV[+] sensory axons were present in control (i.e., *Lrp4*[+/−]) mice at E14.5. They arborized in longitudinal directions upon innervating muscle fibers, a morphology resembling sensory terminals at muscle spindles[32]. There was no detectable difference in PV[+] axon arborization between the two genotypes. Quantitatively, the length of PV[+] terminals was similar between *Lrp4*[−/−] and

control mice (Fig. 3d). Egr3 expression is innervation-dependent, beginning ~E15.5[32,44]; therefore, Egr3 was not detectable in muscle fibers of E14.5 mice (Fig. 3a). Notice that AChR clusters (labeled by α-BTX) were abundant in E14.5 control mice, but not *Lrp4−/−* mice (Fig. 3a), consistent with earlier reports that LRP4 is critical to α-MN NMJ formation. These results suggest that LRP4 may not be required for the navigation of sensory terminals to muscle fibers and initial innervation. At E16.5, Egr3⁺ nuclei in control mice were detected in a segment of intrafusal fibers that were innervated by PV⁺ nerve terminals (Fig. 3b), in agreement with previous reports[44]. Compared with control mice, Egr3 signal was reduced in E16.5 *Lrp4−/−* mice (Fig. 3e); the number of Egr3⁺ nuclei was fewer (Fig. 3f); and the segment of intrafusal fibers with Egr3⁺ nuclei was shorter (Fig. 3g). These phenotypes became more severe as mutant mice developed. PV⁺ terminals were more disorganized and shorter at E18.5 (Fig. 3d) and, in particular,

Egr3 was no longer detectable in fibers in contact with PV+ terminals (Fig. 3c).

In addition, we analyzed the expression of molecules that are implicated in spindle development. As shown in Supplementary Fig. 1b, mRNA levels of *Ig-Nrg1* (*Nrg1-1*, *Nrg1-2*), *Bace1*, and *TrkC* in the DRG and *ErbB2*, *ErbB3*, and *NT3* in muscles were similar between *Lrp4* mutant mice and control mice. *CRD-Nrg1* (*Nrg1-3*) was reduced; however, it is not required for spindle formation[32]. In agreement with histochemical staining, *Egr3* mRNA was reduced in *Lrp4* mutant muscles. These results suggest that LRP4 promotes spindle development by increasing Erg3 expression.

## LRP4 for maintaining muscle spindles

Because *Lrp4* mutant mice die at birth when muscle spindles are not fully developed, we explored the consequences of ablating LRP4 in

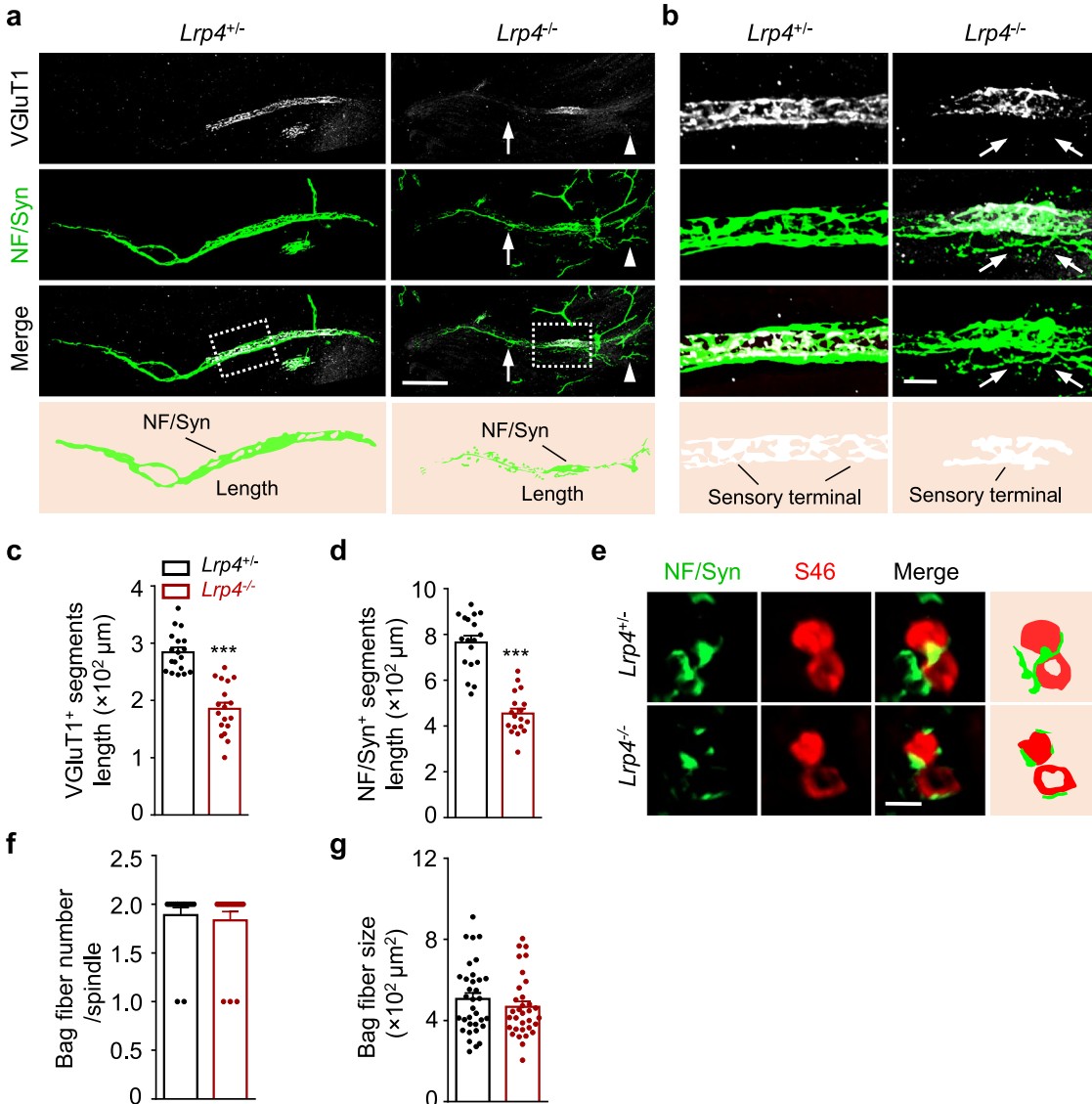

**Fig. 2 | A critical role of LRP4 in spindle development at E18.5. a–d** Disrupted innervation of intrafusal fibers in *Lrp4* mutant. **a** Representative images of TA muscle at E18.5 whole mount stained with VGluT1 (gray) and NF/Syn (green). Arrows, poorly organized axon terminals; arrowheads, axon terminal fragments. Scale bar, 100 μm. **b** Images enlarged from rectangles in (**a**). Disorganized boundaries of NF/Syn⁺ segments by *Lrp4* knockout. Arrows, axons extended into VGluT1 negative region. Scale bar, 20 μm. **c, d** Reduced VGluT1⁺ or NF/Syn⁺ segments length in *Lrp4−/−*. Quantification of VGluT1⁺ segments ($t_{(34)} = 7.386$, ***$p < 0.0001$) or NF/ Syn⁺ ($t_{(34)} = 8.682$, ***$p < 0.0001$) in both groups. $n = 18$ muscle spindles from 3 mice per group, unpaired two-tailed $t$ test. **e** Representative transection images of TA muscle of E18.5 stained with NF/Syn (green) and S46 (Red), and bag fibers (S46⁺) were observed in control and *Lrp4* mutant. Scale bar, 10 μm. Quantification of bag fiber number per spindle (**f**) and bag fiber size (**g**) in both groups. No difference was observed after *Lrp4* knockout. $n = 18$ muscle spindles from 3 mice per group, unpaired two-tailed $t$ test. Data are shown as mean ± SEM. Source data are provided as a Source Data file.

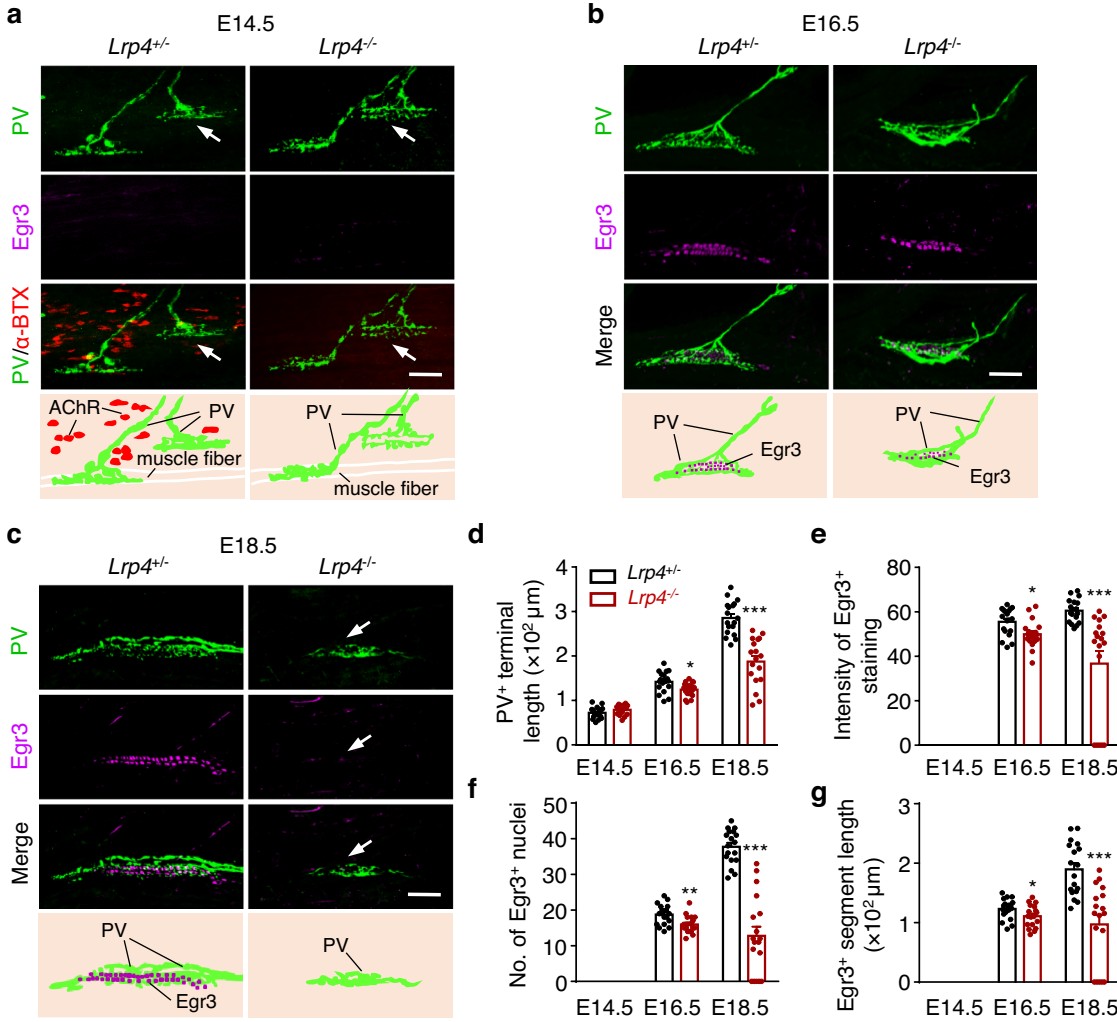

**Fig. 3 | Reduced Egr3 levels in *Lrp4* mutant mice. a** The induction of preliminary muscle spindle in *Lrp4* mutant. Shown were TA muscles at E14.5 in indicated genotypes, whole-mount stained with PV (green), α-BTX (red), and Egr3 (magenta). Arrows, preliminary muscle spindle was induced and innervated by PV⁺ sensory axons. Scale bar, 50 μm. **b, c** Reduced innervation and Egr3 expression in *Lrp4* mutant. Represent images of muscle spindle in TA whole mount stained with PV and Egr3 at E16.5 (**b**) or E18.5 (**c**). Arrows, spindle without Egr3 expression. Scale bar, 50 μm. **d–g** Quantification of data in (**a–c**). **d** Quantification of PV⁺ terminal length (E16.5, $t_{(34)} = 2.638$, *$p = 0.0125$; E18.5, $t_{(34)} = 6.436$, ***$p < 0.0001$). **e** Quantification of intensity of Egr3⁺ nuclei staining (E16.5, $t_{(34)} = 2.708$, *$p = 0.0105$; E18.5, $t_{(34)} = 4.122$, ***$p = 0.0002$). **f** Quantification of number of Egr3⁺ nuclei per spindle (E16.5, $t_{(34)} = 3.146$, **$p = 0.0034$; E18.5, $t_{(34)} = 8.865$, ***$p < 0.0001$). **g** Quantification of Egr3⁺ segment length (E16.5, $t_{(34)} = 2.053$, *$p = 0.0478$; E18.5, $t_{(34)} = 4.879$, ***$p < 0.0001$). $n = 18$ muscle spindles from 3 mice per group, unpaired two-tailed *t* test in (**d–g**). Data are shown as mean ± SEM. Source data are provided as a Source Data file.

adult mice. *Lrp4*^f/f mice were crossed with HSA-Cre^ERT2 mice where CreER is expressed under the promoter of human α-skeletal actin (HSA) that is specific for muscle fibers including intrafusal fibers[23,34,45] (Fig. 4a). Compound *Lrp4*^f/f; HSA-Cre^ERT2 (referred to as imKO) at P30 were treated with 100 mg/kg TAM; 30 days later, LRP4 was almost undetectable (Supplementary Fig. 2a, b). TAM-treated imKO mice started losing weight and muscle strength 25 days after TAM treatment (P55), compared with TAM-treated control mice (Supplementary Fig. 2c, d) and died ~100 days after induction (Supplementary Fig. 2e), in agreement with our earlier report[23].

Muscles in adult mice are large and thick and thus not ideal for whole-mount analysis. Therefore, we studied the extensor digitorum longus (EDL), consisting of four smaller divisions, each of which attaches a specific toe (Supplementary Fig. 3a). EDL division muscles are thin and have been used to visualize both sensory and motor terminals by whole-mount staining[46,47]. Again, NF/Syn antibody was used to visualize axons and nerve terminals and α-BTX to visualize AChRs. In TAM-treated imKO mice, NMJs were fragmented and smaller in size; and AChR intensity was reduced in a time-dependent

manner (Supplementary Fig. 3b–e), consistent with an earlier report[23]. In control mice, intrafusal fibers were coiled by annulospiral sensory endings at the equatorial region (Fig. 4b). In contrast, in TAM-treated imKO mice, sensory nerve terminals were disorganized and fragmented without identifiable annulospinal endings beginning 1 M after TAM treatment (Fig. 4b, arrows). At 2 M after TAM treatment, most axons were degraded, and few were left at the equatorial region (Fig. 4b, arrowheads). Quantitative analysis indicated that the number of spindles with annulospiral endings were reduced to 52.1% and 28% of control, after 1 and 2 M of TAM treatments, respectively (Fig. 4c). Spiral endings (with one circle scored as one) per spindle were decreased from 10.4 in control to 4.81 and 3.25, respectively (Fig. 4d). Annulospiral ending area per spindle (Fig. 4e) and NF/Syn fluorescence intensity (Fig. 4f) were reduced. However, the number of muscle spindles was comparable between TAM-treated control mice and imKO mice (Fig. 4g). Notice that imKO mice die after 2 M of TAM treatment, preventing further time-dependent analysis. These results indicate a role of LRP4 in maintaining the spindles in adult mice.

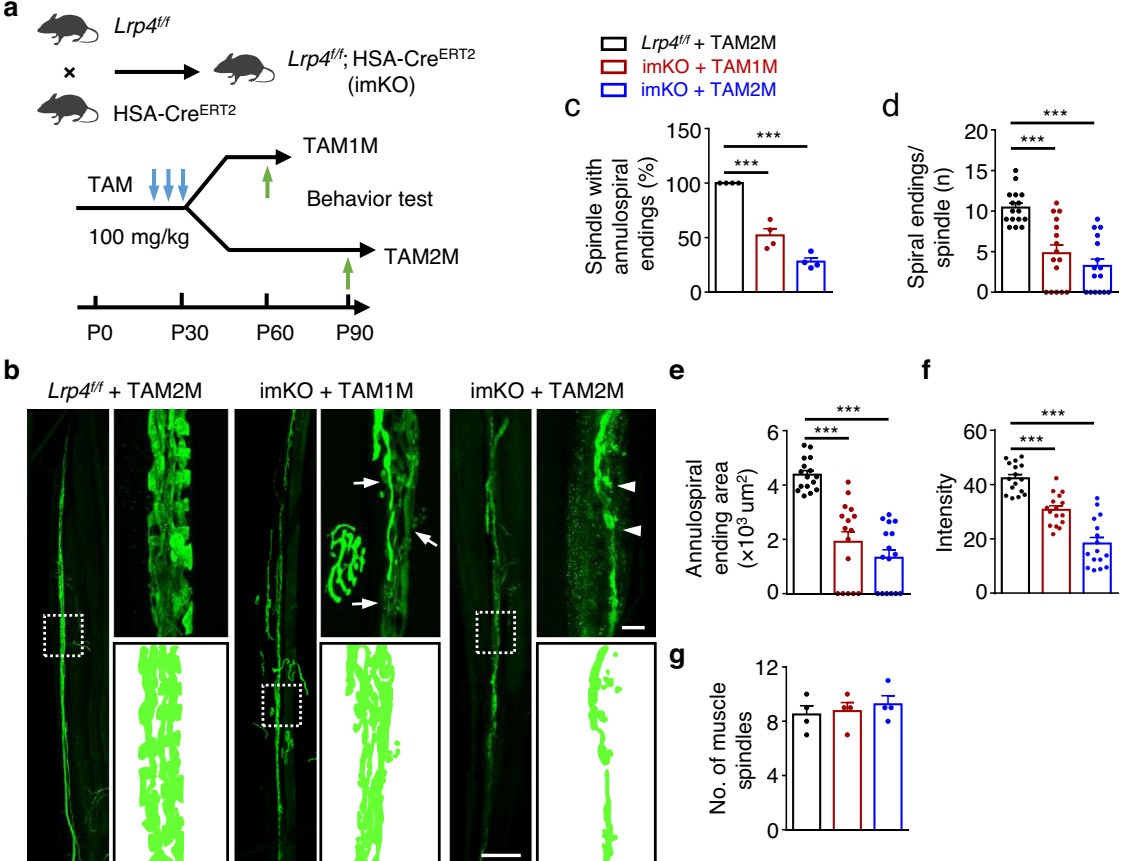

**Fig. 4 | Requirement of LRP4 for maintaining muscle spindles. a** Illustration of breading *Lrp4^f/f*; HSA-Cre^ERT2 (imKO for short) mice and the schematic of tamoxifen administration, behavior tests and tissue collection. **b** Disrupted annulospiral endings in tamoxifen treated imKO mice (P60, P90). Representative images of EDL muscles in indicated groups, whole mount stained with NF/Syn (green) to label axon and axon terminals. Muscle spindle was determined by morphology. Square, images enlarged in right panel. Arrows, axon fragmentations; arrowheads, degenerated axon terminals. Scale bars, 100 μm and 10 μm. **c–g** Quantification of data in (**b**). **c** Reduced percentage of muscle spindles with annulospiral endings in tamoxifen treated imKO mice; $t_{(6)} = 7.893$, ***$p = 0.0002$; $t_{(6)} = 21.614$, ***$p < 0.0001$.

$n = 4$ mice per group, unpaired two-tailed $t$ test. **d** Decreased average spiral endings per muscle spindle ($t_{(30)} = 4.915$, ***$p < 0.0001$; $t_{(30)} = 7.265$, ***$p < 0.0001$). $n = 16$ muscle spindles from 4 mice per group, unpaired two-tailed $t$ test. **e, f** Reduction of annulospiral ending area per muscle spindle ($t_{(30)} = 6.215$, ***$p < 0.0001$; $t_{(30)} = 9.347$, ***$p < 0.0001$) and intensity of muscle spindle in each group ($t_{(30)} = 5.831$, ***$p < 0.0001$; $t_{(30)} = 9.215$, ***$p < 0.0001$). $n = 16$ muscle spindles from 4 mice per group, unpaired two-tailed $t$ test. **g** No difference of total muscle spindle in EDL muscle in each group. $n = 4$ mice per group, unpaired two-tailed $t$ test. Data are shown as mean ± SEM. Source data are provided as a Source Data file.

## Disrupted sensory synapses and proprioceptive sensation

The above results provided morphological evidence for LRP4 in sensory synapse maintenance. We tested this hypothesis by the cholera toxin uptake assay[46]. Once released from the muscle, cholera toxin could bind to its presynaptic receptor and up-taken by presynaptic terminals[48,49]. Cholera toxin uptake assays have been used to study muscle innervation by proprioceptive sensory neurons[46]. We injected cholera toxin subunit B (CTB, 2.5 μl at 0.25 mg/ml) into tibialis anterior (TA)/EDL muscles (Fig. 5a). Five days after injection, lumbar (L) 3 DRG sections were stained for CTB⁺ neurons and PV, a marker of proprioception neurons[46]. DRG neurons for proprioception are usually large in size (1000–2000 μm²) and could be labeled by retrograde CTB injection[46] (Fig. 5b). In fact, almost all large, CTB⁺ neurons in DRG are positive for PV (Fig. 5c). However, fewer such neurons were detected in TAM-treated imKO mice (Fig. 5d, e) although the total numbers of neurons and PV⁺ neurons in DRG were similar between TAM-treated control and imKO mice (Fig. 5f), excluding a potential problem of neuronal death or differentiation. These results suggested that proprioception synapses were impaired by *Lrp4* mutation.

Next, we evaluated the impact of LRP4 inducible deletion on coordinative movement by rotarod test (Fig. 5g). The time when mice were able to remain on an accelerating rod (from 4 to 40 rpm during

5 min) was monitored daily for three days. The times were comparable between control mice and TAM-treated imKO mice 1 M after TAM treatment, but dramatically reduced 2 M after TAM treatment (Fig. 5h). In light that spindles were impaired in mutant mice 1 M after TAM treatment (Fig. 4), we wondered whether rotarod might be a sensitive test for proprioception. Therefore, mice were subjected to balance beam test which reveals coordinative movement[50]. Mice were placed at one end of an elevated beam (6 mm wide x 80 cm long) and monitored for the time to cross the beam leading to a dark box and the number of foot side slips (Fig. 5i). As shown in Fig. 5j, both beam cross time and foot side slips (Fig. 5k) were increased in imKO mice 1 M after TAM, compared to control mice and further increased 2 M after TAM treatment (Fig. 5j, k). These results could suggest that loss of LRP4 in adult mice compromised movement coordination. It is worth noting that α-MN NMJs and muscle strength were also compromised 1 M after TAM treatment (Supplementary Figs. 2 and 3), which may contribute to proprioceptive behavior deficits in rotarod and balance beam tests. We reviewed these caveats in the Discussion.

## Trans LRP4-APP interaction for spindle maintenance

Sensory terminals appeared to be normal in mutant mice lacking agrin or MuSK, suggesting that spindle formation may not require classic

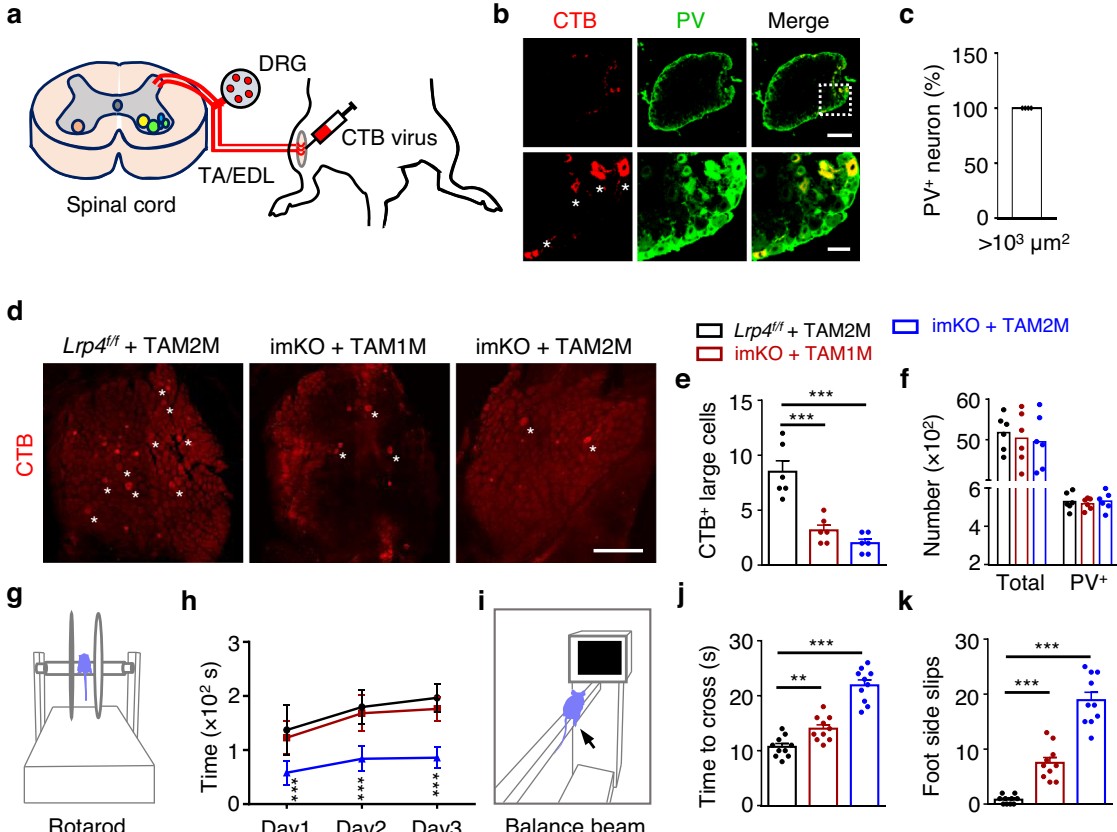

**Fig. 5 | Disrupted sensory synapses and proprioceptive sensation in TAM-imKO mice. a** Schematic diagram of CTB virus injection and labeling. **b** Representative images of DRG sections stained with anti-CTB and anti-PV antibodies. CTB colocalized with PV positive neurons (asterisks). Scale bars, 200 μm and 50 μm. **c** Quantification of the percentage of CTB labeled large neurons (>1000 μm²) with PV staining, $n = 4$ sections from 4 mice (P60). **d** Representative images of DRG at indicated groups labeled by CTB. The positive neurons with soma size larger than 1000 μm² were counted (asterisks). Scale bar, 300 μm. **e** Quantification of CTB labeled large neuron in (**d**); $t_{(10)} = 4.846$, ***$p = 0.0007$; $t_{(10)} = 6.151$, ***$p = 0.0001$, $n = 6$ mice per group, unpaired two-tailed $t$ test. **f** Quantification of total neurons and PV⁺ neurons in each group. $n = 6$ mice per group. **g** Rotarod test. **h** Quantification of the time mice in each group fall off the rotating rod on the first,

second and third testing day. Shorten time on rotating rod in imKO + TAM2M mice; ***$p = 0.0001$ for the first day, ***$p < 0.0001$ for the second and third day. $n = 10$ mice per group, unpaired two-tailed $t$ test. **i** Balance beam walking task. **j** Quantification of the time mice consuming during the beam walking task in each test. imKO mice took longer time during beam walking task two month later after tamoxifen treatment; $t_{(18)} = 3.589$, **$p = 0.0021$; $t_{(18)} = 9.823$, ***$p < 0.0001$. $n = 10$ mice per group, unpaired two-tailed $t$ test. **k** Quantification of the number of side slips during walking on the beam in control or tamoxifen treated mice. Elevated side slips after *Lrp4* knockout; $t_{(18)} = 6.623$, ***$p < 0.0001$; $t_{(18)} = 12.376$, ***$p < 0.0001$. $n = 10$ mice per group, unpaired two-tailed $t$ test. Data are shown as mean ± SEM. Source data are provided as a Source Data file.

agrin signaling[28]. The maintenance of spindles is unlikely to require Erg3 because it is expressed only transiently in developing intrafusal fibers, but not in adult mice[30]. APP is a transmembrane protein that is expressed in sensory neurons. APP levels in DRGs are increased by peripheral nerve injury and return to baseline after recovery[51]. APP could interact in trans with LRP4 and such interaction has been implicated in α-MN NMJ development[52]. Interestingly, APP was located at the annulospiral endings of muscle spindles, supporting a potential involvement in spindles (Fig. 6a, arrows). However, spindles in *App null* mutant mice appeared to be normal (Supplementary Fig. 4a–d). We wonder whether this was due to functional redundancy of APP-like proteins APLP1 and APLP2. APLP2 displays a similar expression pattern to APP[53,54]. Therefore, we suppressed the expression of APP and APLP2 in DRG neurons with AVV-sh*App* and -sh*Aplp2*. Viruses were injected into DRGs at L3-5 that innervate leg muscles. 30 days after injection, many neurons in DRGs expressed mCherry, indicating infection by the AVV virus (Fig. 6b). Quantitative RT-PCR revealed that *App* and *Aplp2* mRNAs were reduced by respective two viruses (Fig. 6c); of them, sh*App*-2 and sh*Aplp2*-1 (referred to as sh*App* and sh*Aplp2*, respectively) appeared to be more efficient (Fig. 6c). Because sensory synapses in muscle spindles are formed by PV⁺ axons[55], we quantified the number

of DRG PV⁺ neurons that were infected by different viruses. As shown in Fig. 6d, e, more than 80% of PV⁺ neurons expressed mCherry.

As shown in Fig. 6f, annulospiral endings of EDL were similar between mice injected with control AAV and those injected with sh*App* or sh*Aplp2*. In contrast, injection of both sh*App* and sh*Aplp2* AAV disrupted the annulospiral structure (Fig. 6f). Quantitatively, the number of spindles with annulospiral endings was decreased to 43.8% in sh*App* + sh*Aplp2*-injected mice, compared with controls ($P < 0.001$, Fig. 6g). Also reduced was the number of spiral endings per spindle from $11.1 \pm 0.53$ in controls to $3.06 \pm 0.67$ of double AVV-injected mice ($P < 0.001$, Fig. 6h). Accordingly, the area of annulospiral endings (Fig. 6i) and the intensity (Fig. 6j) were decreased by knocking down *App* and *Aplp2*. However, the number of spindle remnants in the sh*App* + sh*Aplp2* was comparable to the number of muscle spindle in other groups (Fig. 6k). Taken together, these results suggest that APP/APLP2 of DRG neurons play a role in spindle formation or maintenance.

To determine whether APP/APLP2 act by interacting with LRP4 in intrafusal muscle fibers, HEK293 cells were co-transfected with APP and with Flag-tagged ECD of LRP4. Cell lysates were incubated with

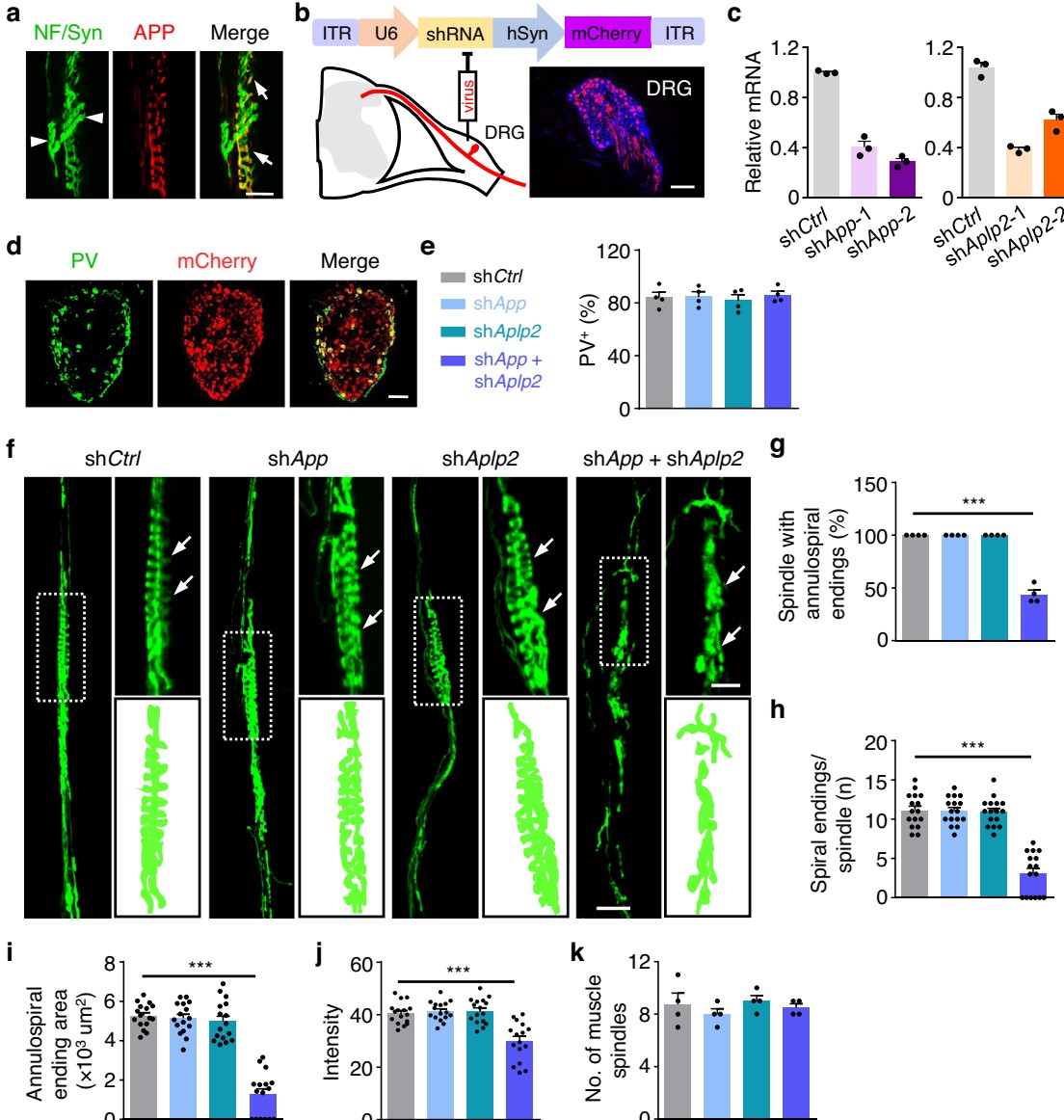

**Fig. 6 | Degenerated annulospiral endings after *App* and *Aplp2* knocked out.**
**a** EDL muscle of WT mouse was whole mount stained with NF/Syn (green) and APP (red). Arrows, APP and NF/Syn positive annulospiral endings; Arrowheads, only NF/Syn positive axons. Scale bar, 20 μm. **b** Construction and expression of AAV shRNA in DRG (P60). Scale bar, 100 μm. **c** Quantitative RT-PCR was performed to test the silence efficiency of *App* shRNAs and *Aplp2* shRNAs one month after the injection. sh*App*-1 and sh*App*-2 were reduced to 0.405 ± 0.045 and 0.287 ± 0.026, respectively. sh*Aplp2*-1 and sh*Aplp2*-2 were decreased to 0.387 ± 0.016 and 0.62 ± 0.048, respectively. Data are shown as mean ± SEM, *n* = 3 mice per group. **d** Representative images of virus infected (red) DRG section stained with anti-PV (green). Scale bar, 100 μm. **e** Quantification of the percentage of PV⁺ neurons infected by mCherry-expressing virus. Data are shown as mean ± SEM, *n* = 4 mice per group. **f** Representative images of muscle spindle labeled by NF/Syn in sh*Ctrl*, sh*App* alone,

sh*Aplp2* alone and sh*App* plus sh*Aplp2* groups. Arrows, annulospiral endings, rectangles, images enlarged in the right panel. Scale bars, 50 μm and 20 μm. **g–k** Quantification of data in (**f**). **g** Reduced percentage of muscle spindle with annulospiral endings in sh*App* + sh*Aplp2*; $F_{(3, 12)}$ = 174.2, ***$p < 0.0001$. $n = 4$ mice per group, one-way ANOVA. **h** Less spiral endings per spindle by the silence of *App* and *Aplp2*; $F_{(3, 60)}$ = 57.34, ***$p < 0.0001$. **i** Decreased average area of annulospiral ending per muscle spindle in sh*App* + sh*Aplp2* group; $F_{(3, 60)}$ = 72.03, ***$p < 0.0001$. **j** Reduced intensity per muscle spindle after the silence of *App* and *Aplp2*; $F_{(3, 60)}$ = 18.78, ***$p < 0.0001$. $n = 16$ muscle spindles from 4 mice per group, one-way ANOVA in (**h–j**). **k** No comparable difference in muscle spindle number among groups. $n = 4$ mice, one-way ANOVA. Data are shown as mean ± SEM. Source data are provided as a Source Data file.

anti-Flag antibody immobilized on beads and resulting immunocomplexes were probed with anti-APP antibody. As shown in Fig. 7a (lane 1), APP was detected in the complex with LRP4 ECD, indicating that the two proteins may interact, in support of the model. Next, we determined which motifs of LRP4 ECD is critical for binding to APP by co-transfected with LRP4 ECD deletion mutants. As shown in Fig. 7a (lane 2), deleting low-density lipoprotein class A repeats (LDLa) diminished the APP interaction while ECD without other domains had no effect. This result identified LDLa as a critical motif for APP interaction. On the

other hand, deleting the E1 motif in APP abolished the interaction with LRP4 (Fig. 7b), suggesting an essential involvement of E1, in agreement with a previous report[52].

If spindle formation or maintenance requires the APP-LRP4 interaction, disrupting the interaction should impair the spindles. To test this, recombinant E1 domain and LDLa were purified (Supplementary Fig. 4e) and injected into TA/EDL of 2 M-old mice (Fig. 7c). The E2 domain did not interact with LRP4 and thus used as control. Muscle spindles were impaired in E1- and

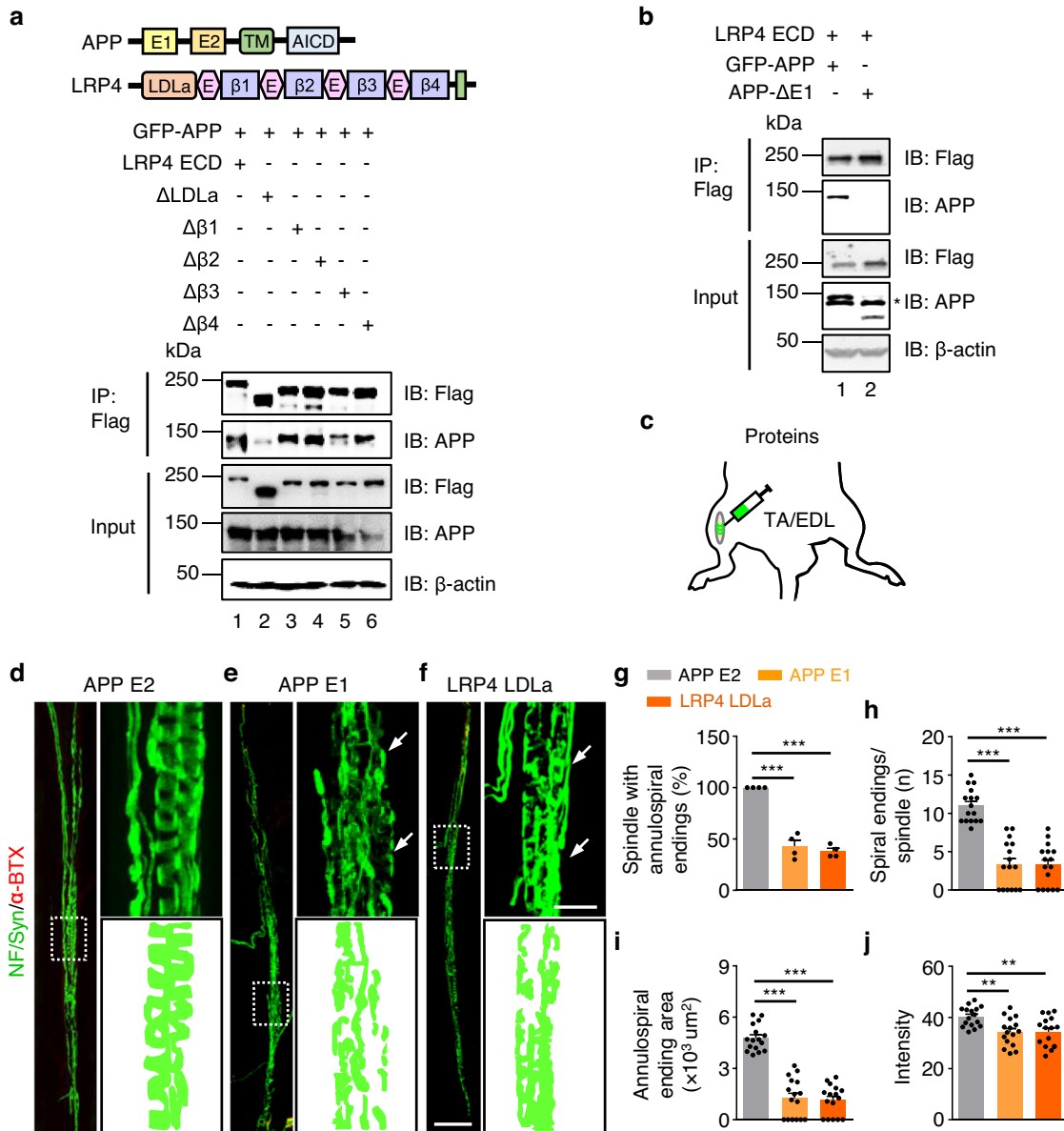

**Fig. 7 | LRP4-APP trans interaction for muscle spindle structure.**
**a**, **b** Requirement of APP E1 and LRP4 LDLa for binding. **a** co-IP of GFP-APP co-transfected with Flag labeled LRP4 ECD, ECD deleted with LDLa (ΔLDLa), Δβ1, Δβ2, Δβ3 and Δβ4. This experiment was repeated three times independently with similar results. **b** co-IP of LRP ECD co-transfected with GFP labeled APP or APP ΔE1. Asterisks, non-specificity bands. Uncropped blots in Source Data. This experiment was repeated three times independently with similar results. **c** Schematic diagram of recombinant proteins injection (P60). **d–f** Impaired annulospiral endings after the injection of recombinant APP E1 and LRP4 LDLa (P60). EDL muscles injected by APP E2 (**d**), APP E1 (**e**) and LRP4 LDLa (**f**) recombinant proteins were whole mount stained with NF/Syn (green) and α-BTX (red). Arrows, the equatorial region of

muscle spindle; rectangles, enlarged regions at side. Scale bars, 100 μm and 20 μm. **g–j** Quantification of data in (**d–f**), APP E2 serves as control. **g** Reduced percentage of muscle spindle with annulospiral endings by APP E1 and LRP4 LDLa injection; $t_{(6)} = 9.642$, ***$p < 0.0001$; $t_{(6)} = 21.139$, ***$p < 0.0001$. $n = 4$ mice per group, unpaired two-tailed $t$ test. **h** Less spiral endings per spindle after APP E1 and LRP4 LDLa injection; $t_{(30)} = 8.152$, ***$p < 0.0001$; $t_{(30)} = 8.924$, ***$p < 0.0001$. **i** Decreased innervation area of muscle spindle in APP E1 and LRP4 LDLa injected mice; $t_{(30)} = 10.108$, ***$p < 0.0001$; $t_{(30)} = 12.279$, ***$p < 0.0001$. **j** Reduced intensity after the injection of APP E1 and LRP4 LDLa; $t_{(30)} = 3.589$, **$p = 0.0012$; $t_{(30)} = 3.398$, **$p = 0.0019$. $n = 16$ muscle spindles from 4 mice, unpaired two-tailed $t$ test in (**h–j**). Data are shown as mean ± SEM. Source data are provided as a Source Data file.

LDLa-injected groups, compared with control (Fig. 7d–f). Though the number of spindle remnants or muscle spindles were comparable (data not shown), the spindles with annulospiral endings were reduced to 43% and 38% by E1 and LDLa, respectively ($P < 0.001$, Fig. 7g). The number of spiral endings per muscle spindle were reduced from 11 to 3.31 and 3.25 ($P < 0.001$, Fig. 7h), respectively. Besides, the annulospiral ending area (Fig. 7i) and the intensity (Fig. 7j) of muscle spindle were reduced by E1 and LDLa. These results support the hypothesis that the LRP4-APP/APLP2 interaction is involved in the maintenance of muscle spindles.

## Restoring spindle structure and function in aged mice by LRP4 expression

Ageing is often associated with muscle weakness as well as reduced ability to coordinate movement[56–58]. For example, 12 M mice showed more side slips and took longer to cross the beam in the balance beam test, compared with 3 M mice (Supplementary Fig. 5a, b). These deficits worsened at 28 M (Supplementary Fig. 5a, b). These results suggest impaired movement coordination in aged mice, in agreement with previous reports[59]. For reasons not well understood, however, the running time on the rotorad test was similar between 3 M mice and

aged mice (both 12 and 28 M)[59] (Supplementary Fig. 5c). The poor performance in the balance beam test was associate with a reduction in CTB uptake by DRG neurons (Supplementary Fig. 5d, e), suggesting proprioception synapse impairment. α-MN NMJs were fragmented and AChR intensity reduced, and the endplate area per NMJ was also decreased in aged mice (Supplementary Fig. 5f–i), in agreement with previous reports[24,60–62]. Remarkably, annulospiral endings at the equatorial region of intrafusal fibers were disorganized, compared with those in 3 M mice (Supplementary Fig. 6a, b, arrows). Axons were degenerated with bleb-containing endings in aged mice (Supplementary Fig. 6a, c, arrowheads). The width of annulospiral endings was reduced and the inter-rotational distance (IRD, the distance between spirals) were increased in 28 M mice (Supplementary Fig. 6d–f). These results were consistent with the literature[63,64] and indicated that the compromised coordinative movement in aged mice correlated with altered muscle spindles.

As LRP4 was required for the maintenance of muscle spindle, we wondered whether LRP4 levels were reduced in aged spindles. To this end, we labeled monoclonal anti-LRP4 antibody with red fluorescent dye CF555 (referred to as 555-anti-LRP4) and co-stained muscles with 555-anti-LRP4, anti-NF/Syn antibody and 488-α-BTX. LRP4 was localized at α-MN NMJs in control mice, but absent from fragmented α-MN NMJs in TAM-treated imKO mice (Supplementary Fig. 7a). It was not detected in deformed muscle spindle of TAM-treated imKO mice (Supplementary Fig. 7b). Consistent with early publication[28], LRP4 was abundantly located at annulospiral endings of muscle spindle in the 3 M group (Fig. 8a). In the 12 M group, LRP4 was co-localized with most, but not all annulosprial endings (Fig. 8a, arrows). At 24 M, LRP4 in muscle spindles was significantly reduced and seldom co-localized with annulospiral endings (Fig. 8a, arrows). The relative intensity of LRP4 was reduced in aged muscle spindles (Fig. 8b). Quantitative RT-PCR showed that, compared with 3 M, *Lrp4* mRNA was reduced in the muscle spindle of 12 M and 24 M mice (Fig. 8c).

If LRP4 reduction contributes to spindle deficits in aged mice, increasing LRP4 levels might protect spindles from deterioration during ageing. To test this, we evaluated the coordinative movement in aged *Flag-Lrp4* transgenic mice that express Flag-LRP4 in skeletal muscles including intrafusal fibers[24] (Fig. 8d). The longest running time on rotating rod was comparable between control and *Flag-Lrp4* mice (Fig. 8e). However, compared with the control mice, 24 M *Flag-Lrp4* mice showed reduced side slips (Fig. 8f) and spent less time to cross the balance beam (Fig. 8g). Muscle spindles in control mice exhibited blebs (Fig. 8h, arrowhead), unraveling and deformation of annulospiral endings. However, these phenotypes appeared to be less severe in *Flag-Lrp4* mice (Fig. 8h–j, arrows). Axon size (Fig. 8k) and IRD (Fig. 8l) were improved in *Flag-Lrp4* mice. Taken together, these results suggest that increasing LRP4 in aged mice could alleviate spindle deficits and improve coordinative movement.

## Discussion

In this study, we provide evidence that LRP4 is expressed abundantly in intrafusal muscle fibers and is required for the formation and maintenance of muscle spindles. In *Lrp4* mutant mice, sensory nerve terminals were disorganized, discontinuous or fragmented, indicative of a role of LRP4 in sensory terminal development. Moreover, induced deletion of LRP4 in adulthood caused fragmentation of sensory nerve terminals, diminished annulospinal endings, and impaired proprioceptive behaviors, demonstrating that LRP4 is required for spindle maintenance. Furthermore, spindles disintegrated in aged mice, reducing movement coordination that was associated with a reduction in LRP4 in spindles. Transgenic expression to increase LRP4 levels improved both spindle sensory endings and coordinative movement in aged mice. These results uncover a mechanism of spindle development and maintenance.

LRP4 is a transmembrane protein with a large ECD and a small intracellular domain (ICD) that is expressed in many tissues including muscles, brain, kidney and bone[14]. We showed previously that the soluble ECD is able to function as a receptor (albeit with lower efficacy than the full length LRP4) for agrin to activate MuSK[65]. In accord, mutant mice or bovine expressing the ECD without the transmembrane domain or ICD are able to survive but display syndactyly henotypes[66,67], suggesting the ICD may be necessary for signaling to prevent mulefoot or syndactyly. Most *Lrp4* mutations in patients with Cenani-Lenz syndrome (CLS) are recessive and believed to alter its expression or function[68–71]. In mice, neonatal lethality can be caused by null mutation or mutations missing a critical region in the ECD such as *mitt* and *mte*[18]. Several mutations have been identified in human that produce truncated LRP4 mutant proteins that lack a segment of the ECD (c.2401 A > T, c.3062 del C, and c.199–200ins GATTCAG)[70] or the transmembrane domain (c.4952–4987 del)[68]. Although none of them were homozygous, however compound heterozygous alterations (c.2401 A > T and c.3062 del C or c.4952–4987 del and c.199–200ins GATTCAG) caused prenatal lethality[68,70]. A single amino acid missense mutation in the first β1 propeller domain (c.1585 G > A) causes neonatal lethality[68]. These results have the following implications with an assumption that the mutant proteins are expressed in a manner dependent on gene-dosage. First, compound, severe loss-of-function *Lrp4* mutations cause pre- or neonatal lethality, in agreement with mouse studies. Second, LRP4 ECD is critical for its function. Third, the lethality of the compound heterozygous mutations (c.4952–4987 del and c.199–200ins GATTCAG) suggest that a single copy of the ECD is not sufficient for signal transduction. Alternatively, the ECD truncation mutant protein produced by c.199–200ins GATTCAG could be a dominant negative. However, homozygous mutation of c.289 G > T that leads to a premature stop codon at amino acid 97 (p.E97X) at the very beginning of the large ECD was not lethal[72], which remains to be a puzzle difficult to explain.

At the α-MN NMJ, genes that encode proteins necessary for its structure and function are transcribed specifically in sub-synaptic nuclei[1,2,73]. Hence, the promoter activity of many of these genes are locally active including that of *Lrp4*[39]. We show here that *Lrp4* mRNA and promoter activity are high in intrafusal muscle fibers, compared with regular muscle fibers (Fig. 1). A recent single nuclei RNA-sequence analysis revealed *Lrp4* mRNA in the nuclei close to the γ-MN NMJ in intrafusal muscle fiber[74]. It was not detected in the nuclei associated with the annulospiral endings, probably due to the low detection sequencing efficiency (1000–2000 mRNAs per nucleus); mRNAs of other NMJ markers such as *MuSK*, *DOK7* or *rapsyn* were not detectable in γ-MN NMJ-associated nuclei[74]. Future, more efficient studies are warranted to determine detailed gene profiles of nuclei close to annulospiral endings.

Assessment of proprioceptive behavior requires muscle strength. Impaired proprioception has been documented as a secondary effect in disorders that impinge on α-MN NMJs such as myasthenia gravis[75,76]. Nevertheless, proprioceptive behaviors were reported normal in muscle dystrophy mouse models with mild muscle weakness and bodyweight loss[77]. Evidently, LRP4 is critical to the formation and maintenance of α-MN NMJs[18,23]. *Lrp4 null* mutation impaired development of sensory terminals (Figs. 2, 3); because of the neonatal lethality, it prevents behavioral studies of proprioceptive function. However, proprioceptive behaviors were compromised in *Lrp4* conditional mutant mice (Fig. 5), associated with severe deficits in spindle morphology and sensory synapses (Fig. 4). A parsimonious interpretation of these results supports a role of LRP4 in spindle formation and maintenance.

Ig-Nrg1 produced by DRG neurons is a master regulator for spindle formation[32–34]. By activating ErbB2/3 receptor tyrosine kinases, Ig-Nrg1 promotes the expression of Egr3 that is necessary for induction of intrafusal muscle fibers and for spindle formation. *Lrp4* mutation

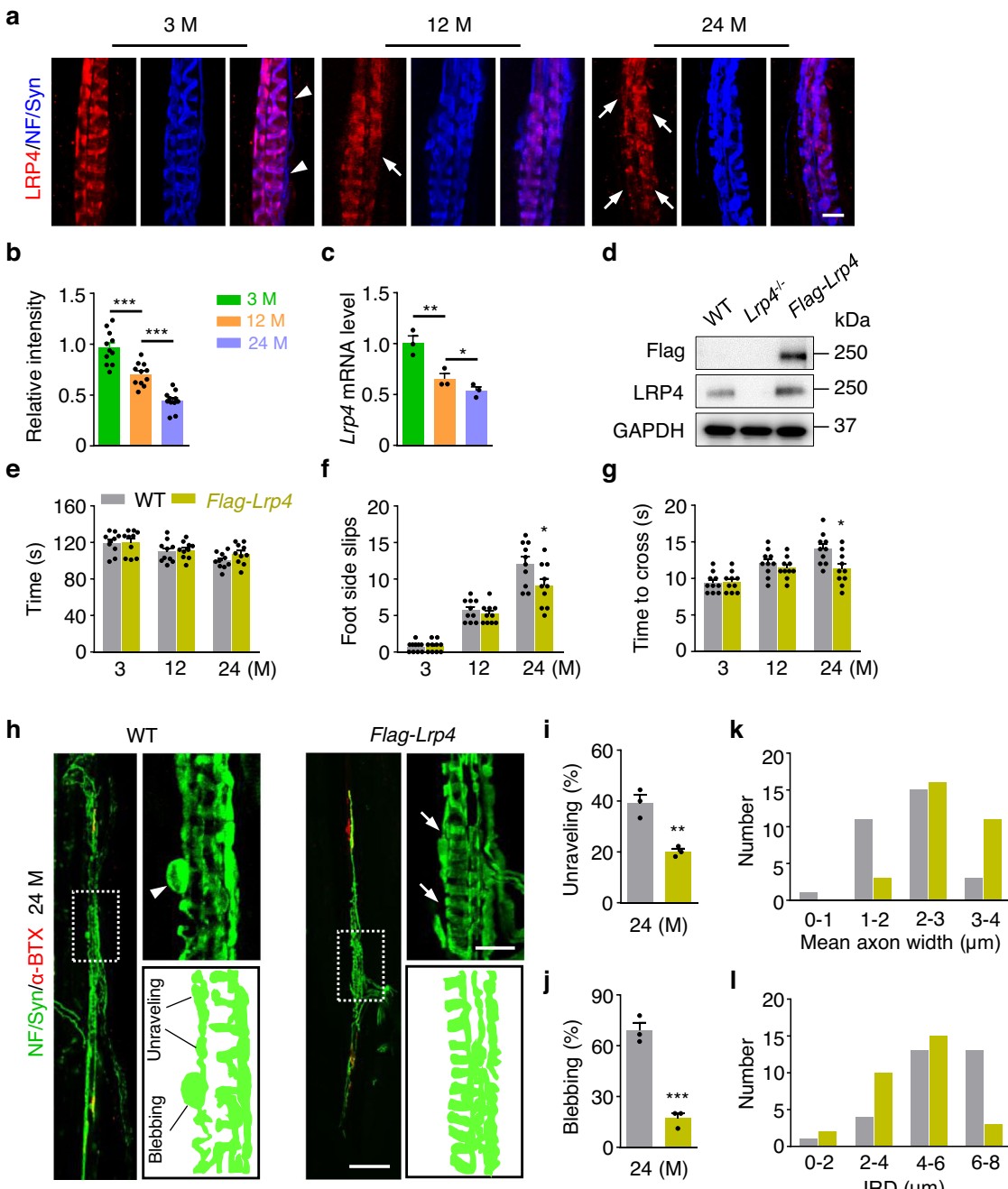

**Fig. 8 | Restoring spindle structure and function in aged mice by LRP4 expression. a** Representative images of EDL muscles of 3 M, 12 M, and 24 M-old mice, whole mount stained with 555-anti-LRP4 (red) and NF/Syn (blue). Arrowheads, non-annulospiral axons had no LRP4 expression. Arrows, LRP4 negative annulospiral endings in aged mice. Scale bar, 10 μm. **b** Quantification of the relative intensity of LRP4; $t_{(20)} = 9.063$, ***$p < 0.0001$; $t_{(20)} = 4.34$, ***$p = 0.0003$. $n = 11$ muscle spindles from 3 mice per group, unpaired two-tailed $t$ test. **c** Quantitative RT-PCR tested the relative expression of *Lrp4* mRNA in 3 M, 12 M or 24 M mice; $t_{(4)} = 5.861$, **$p = 0.004$; $t_{(4)} = 4.006$, *$p = 0.016$. $n = 3$ mice per group, unpaired two-tailed $t$ test. **d** Expression of Flag-LRP4 in skeletal muscle was confirmed by Western blotting. Uncropped blots in Source Data. **e** Quantification of longest running time on rotating rod between WT and *Flag-Lrp4* mice at indicated age. **f** Quantification of

side slips in beam walking task of WT and *Flag-Lrp4* mice at indicated age; $t_{(18)} = 2.251$, *$p = 0.037$. **g** Quantification of time required for beam walking task; $t_{(18)} = 2.793$, *$p = 0.012$. $n = 10$ mice per group, unpaired two-tailed $t$ test in (**e**–**g**). **h** Representative images of muscle spindle in EDL muscles of 24 M-old mice, whole mount stained with NF/Syn (green) and α-BTX (red). Arrowhead, bleb of axon; arrows, annulospiral endings. Scale bars, 100 μm and 20 μm. **i** Quantification of unraveling muscle spindle in each group; $t_{(4)} = 5.584$, **$p = 0.005$. **j** Quantification of the incidence of axon with large blebs in each group; $t_{(4)} = 9.55$, ***$p = 0.0007$. $n = 3$ mice per group, unpaired two-tailed $t$ test in (**i**) and (**j**). The distribution of mean axon width (**k**) and inter-rotational distance (**l**) in aged WT and *Flag-Lrp4* mice. Data are shown as mean ± SEM, $n = 3$ mice per group. Source data are provided as a Source Data file.

apparently had little effect on S46$^+$ intrafusal fibers (Fig. 2e–g); spindles at E14.5 were similar between *Lrp4* mutant and control mice (Fig. 3a, d–g), suggesting that LRP4 is not necessary for the initial steps of spindle formation. However, the maturation of spindles appeared to

be stalled in *Lrp4* mutant mice after E16.5 (Fig. 3b–g). *Lrp4* mutation has little effect on mRNA levels of *Ig-Nrg1* and *Bace1* in DRG and of *ErbB2* and *ErbB3* in muscles (Supplementary Fig. 1b), suggesting the Ig-Nrg1-ErbB2/3 signaling may not be compromised by *Lrp4* mutation.

No change was observed in mRNA levels of *NT-3* or *TrkC*, suggesting normal retrograde signaling. However, Egr3 is reduced at both mRNA and protein levels in *Lrp4* mutant mice (Supplementary Fig. 1b, Fig. 3c), revealing a mechanism for LRP4-dependent spindle development. These results also suggest that the expression of this transcription factor requires LRP4.

Erg3 is an immediate early gene that has been implicated in a plethora of biological events ranging from tumor development to allergy and from drug abuse to psychiatric disorders such as schizophrenia[78–81]. Its expression is likely regulated by multiple pathways although underlying mechanisms are unclear. Agrin was reported to increase Egr3 expression in cultured myotubes[82]; however, *agrin* or *MuSK* mutation does not alter muscle spindle sensory innervation[28]. Egr3 levels were increased by VEGF in human umbilical vein endothelial cells[83], by BDNF in primary hippocampal neurons[84] and by transforming growth factor-β in skin fibroblasts[85]. With a large extracellular domain, LRP4 may serve as a receptor for many ligands; in addition to agrin, they include Wnt and the negative regulators of Wnt signaling Dickkopf-1 (DKK1) among others such as sclerostin, Gremlin1, Wise, APP and ApoE[86,87]. We showed that LRP4 is required for Wnts to regulate AChR clustering[88]. In addition, LRP4 controls bone homeostasis by regulating Wnt signaling[89]. β-catenin, a major mediator of Wnt canonical pathway, promotes Egr3 expression[90,91]. Wnt signaling also regulates the expression of Erg3 by suppressing the negative regulators of Wnt signaling DKK1[92]. Interestingly, Wnt7a is selectively expressed in γ-MNs in contrast to α-MNs and thus has been used as an embryonic marker of γ-MNs[93]. *Wnt7a* mutation reduced the motor neuron pool, but seemed to have little effect on the innervation of intrafusal muscle spindles[93]. This result is not unexpected because motor neurons express many isoforms of Wnts[94]. The Wnt receptor Frizzled 5 is expressed in muscle spindle capsules[93]. However, LRP4 could sequester LRP5/6-mediated Wnt signaling in vitro[66], suggesting a complex role of Wnts. It is worthy determining whether and how Wnt signaling regulates spindle formation.

A continuous contact between sensory terminals and intrafusal fibers is important for the maintenance of muscle spindles in adults. For example, denervation causes deterioration and eventual loss of muscle spindles[95]. In addition to diffusible signals discussed above, LRP4 regulation of spindles may be mediated by its in-trans interactions with transmembrane proteins at sensory terminals. Such a mechanism may contribute to the formation of α-MN NMJs although the identity of the presynaptic binding protein remains unclear[65,96]. Here we revealed an in-trans mechanism for spindle maintenance, i.e., by interacting with APP that is expressed in sensory terminals. First, APP is expressed in DRG neurons and its levels are increased by peripheral nerve injury and return to baseline after recovery[51]. Second, APP was located at the annulospiral endings of muscle spindles (Fig. 6a). Third, the annulospiral structure was disrupted in mice whose expression of both APP and APLP2 (an APP isoform) in DRG neurons was suppressed (Fig. 6f). Fourth, APP and LRP4 are known to interact in trans[52]; we showed that such interaction requires the LDLa domain in LRP4 and the E1 motif in APP (Fig. 7a, b). Finally, and importantly, disrupting the LRP4-APP interaction by overexpressing LDLa or E1 impaired muscle spindles in mice (Fig. 7d–f). These results support the hypothesis that LRP4-APP in-trans interaction is critical to the maintenance of muscle spindles.

Ageing alters the structure and function of muscle spindles, resulting in a decline in proprioceptive sensation in humans and rodents[97,98]. We showed that in aged mice sensory terminals lost typical annulospiral configuration and became irregular in shape in agreement with earlier observations in aged rats[63]. Conditional deletion of LRP4 in adult mice impaired muscle spindle annulospiral endings, indicating a critical role of LRP4 in maintaining muscle spindles (Fig. 4). This finding, together with reduced levels of LRP4 in aged muscles[24], suggests a potential pathological mechanism for impaired structure and function of muscle spindles in aged mice. This hypothesis is further supported by improved sensory terminal morphology and coordinative movement by increasing expression of LRP4 (via crossing *Flag-Lrp4* transgenic mice) (Fig. 8). Notice that LRP4 in *Flag-Lrp4* transgenic mice is not only expressed in intrafusal fibers but also extrafusal fibers; therefore, the improvement in coordinative movement may involve improved α-MN NMJs[24].

## Methods

### Animals

All animal experiments in this study were carried out according to protocols approved by the Institutional Animal Care and Use Committee of Case Western Reserve University and performed in compliance with the National Institutes of Health Guide for the Care and Use of Laboratory Animals. *Lrp4* Cre[ERT2] mice carry a 3′ FRT flanked neo cassette and P2A-GFP-CreERT2 cassette inserted to exon 38[38]; *Lrp4* LacZ reporter mice with a lacZ cassette driven by the promotor of the *Lrp4* gene were from KOMP (VG15248)[40]; Exon 1 of the *Lrp4* gene in *Lrp4*[f/f] mice were flanked by loxP sites[65]; HSA-Cre[ERT2] mice express Cre[ERT2] selectively in skeletal muscles[99]; and *Flag-Lrp4* transgenic mice express *Flag-Lrp4* cDNA under human α-skeletal actin promoter[24]. Older mice (12 M, 24 M, 28 M-old) were acquired from the National Institute on Ageing[100]. Ai9 (Stock No: 007905) and *App null* transgenic mice (Stock No: 004133) were purchased from Jackson's laboratory. Mice were backcrossed into C57BL/6 J background and housed no more than five per cage in a room with a 12 h light/dark cycle with *ad libitum* access to water and rodent chow diet (Diet 7097, Harlan Teklad). Mice were kept at an ambient temperature of 23 °C and humidity of 40–60%. 10 mg/ml tamoxifen (Sigma-Aldrich, Cat# T5648) was prepared in corn oil (Sigma-Aldrich, Cat# C8267) mixed with ethanol (9:1 ratio). Mice were injected with 100 mg/kg tamoxifen every other day (i.p., three times)[40]. Both sexes were used in the study and mouse ages were described in figure legends. Experimenters were blinded to genotypes and treatments.

### Reagents and antibodies

Chemicals, unless other indicated, were purchased from Sigma-Aldrich. The information of primary antibodies used was as follows: CF568-labeled α-bungarotoxin (α-BTX, Biotium Cat# 0006, 1:1000 for staining); CF488A-labeled α-bungarotoxin (488-α-BTX, Biotium Cat# 0005, 1:1000 for staining); rabbit anti-neurofilament (NF, Cell Signaling Technology Cat# C28E10, 1:1000 for staining); rabbit anti-synapsin (Syn, Cell Signaling Technology Cat# D12G5, 1:1000 for staining); mouse anti-myosin heavy chain (Developmental Studies Hybridoma Bank Cat#S46, 1:200 for staining); mouse anti-Egr3 (Santa Cruz Cat# sc-390967, 1:500–1000 for staining); rabbit anti-Parvalbumin (Swant Cat#PV25, 1:1000 for staining); guinea pig anti-VGluT1 (Sigma-Aldrich Cat# AB5905, 1:1000 for staining); goat anti-cholera toxin subunit B (CTB) (Sigma-Aldrich Cat#227040, 1:1000 for staining); mouse anti-LRP4 (UC Davis/ NIH NeuroMab Facility Cat# 75–221, 1:1000 for WB, 1:200 for staining); mouse anti-APP (6E10) (BioLegend Cat# 803001, 1:1000 for staining); rabbit anti-APP (Sigma-Aldrich Cat# A8717, 1:1000 for WB); rabbit anti-Flag (Sigma-Aldrich Cat# F7425, 1:2000 for WB); mouse anti-GAPDH (Novus Biologicals Cat# NB 600-502, 1:3000 for WB); mouse anti-β-actin (Cell Signaling Technology Cat# 3700, 1:5000 for WB). The information of secondary antibodies used was as follows: Alexa Fluor 488 donkey anti-rabbit IgG (Cat# 711-547-003); Alexa Fluor 647 donkey anti-guinea pig (Cat# 706-605-148); Alexa Fluor 647 donkey anti-rabbit (Cat# 711-605-152); Alexa Fluor 594 donkey anti-mouse (Cat# 715-585-150); Alexa Fluor 594 donkey anti-Goat (Cat# 705-585-003) were all diluted by 5% goat serum and 5% BSA in PBS (1:1000) and purchased from the Jackson ImmunoResearch. Horseradish peroxidase (HRP)-conjugated goat anti-rabbit IgG (Cat# 32260) and goat anti-mouse IgG antibodies (Cat# 32230) were from Thermo Fisher Scientific and used at 1:4000 for Western blot.

## Immunofluorescence

Muscles were whole-mount stained[24,46]. For embryonic mice, limbs or diaphragms were isolated and fixed in 4% paraformaldehyde (PFA) in PBS at 4 °C overnight, rinsed with 0.1 M glycine in PBS for 30 min and blocked for 2 h with the blocking buffer containing 2% Triton X-100 in 5% BSA and 5% goat serum. After that, muscles were incubated with primary antibody in blocking buffer at 4 °C overnight, washed with 0.5% Triton X-100 in PBS three times for 10 min each, and then incubated with secondary antibodies in blocking buffer overnight at 4 °C. The samples were washed with 0.5% Triton X-100 in PBS and mounted with Vectashield (H1200) mounting medium and coverslip. Muscle spindles were determined by staining for PV or VGluT1 (two markers for embryonic sensory nerve terminals) or co-stained with NF/Syn antibodies. Muscle spindles were distinguished from Golgi tendon organs which have highly branched endings[101].

Leg muscles were fixed as described above and EDL was isolated. Muscle spindles were morphologically identified with annulospiral endings at equatorial region by staining with NF/Syn antibodies[35]. Notice that in adult mice, Golgi tendon organs are concentrated at the end of muscle fibers.

For muscle cross-section staining, muscles were fixed in 4% PFA in PBS at 4 °C overnight, and dehydrated in 30% sucrose at 4 °C overnight. The muscles were frozen at −80 °C in Cryo-embedding medium and cut into 14 μm sections on a cryostat at −25 °C. Sections were rinsed with PBS and incubated with blocking buffer (0.5% Triton X-100 in 3% BSA and 3% goat serum) for 2 h at room temperature and then primary antibodies in blocking buffer at 4 °C overnight. The samples were washed with 0.5% Triton X-100 three times for 10 min each and incubated with secondary antibodies in blocking buffer at 4 °C overnight. After washing, the samples were mounted with Vectashield mounting medium and coverslip. For immunostaining of PV and S46 in DRG and transection spindle slices, citrate buffer antigen retrieval (10 mM, pH = 6.0, 95 °C, 20 min) was performed. For transection view, muscle spindles were determined by morphology characters: encapsulated structures, clustered nucleus, periaxial space of Sherrington, and intrafusal fibers[35].

## Quantitative reverse transcription-PCR analysis (RT-qPCR)

Total RNA was extracted with Trizol (Invitrogen Cat# 15596-026) and reverse-transcribed to cDNA (Promega Cat# A2801). Quantitative RT-PCR was performed with Step-One Plus system (Thermo Fisher Scientific Cat# 4376600) using SYBR GreenER qPCR Mix with gene-specific primers. RNA levels were normalized to internal control (*Gapdh*) and the primers were as follows: *Lrp4* (F: 5′-AGTCA CCGCA AGGCT GTCAT TA-3′, R: 5′-GTTGG CACTA TTGAT GCTCT TGG-3′); *Nrg1-1* (F: 5′- TCATC TTTAG CGAGA TGTCT G-3′, R: 5′- CTCCT GGCTT TTCAT CTCTT TCA-3′); *Nrg1-2* (F: 5′- GAGAC TGGCC GCAAC CTCA-3′, R: 5′- TGACT CCTGG CTTTT CATCT CTTT-3′); *Nrg1-3* (F: 5′- GGACC CCTGA GGTGA GAACA-3′, R: 5′- CAGTC GTGGA TGTAG ATGTG G-3′); *Bace1* (F: 5′- TGCTG CCATC ACTGA ATCGG AC-3′, R: 5′- GGAAT GTGGG TCTGC TTCAC CA-3′); *TrkC* (F: 5′- GTCTG ATGCG AGCCC TACAC C-3′, R: 5′- AGAGA ACCAC CAGAA GGACG CA-3′); *ErbB2* (F: 5′- GACCT CAGTG TCTTC CAGAA CC-3′, R: 5′- TGCGG TGAAT GAGAG CCAAT CC-3′); *ErbB3* (F: 5′- AGGCT CATTG CTTCT CCTGC CA -3′, R: 5′- GAAAA TGGGC GCATC GAGCA CA -3′); *Egr3* (F: 5′- CTGAC AATCT GTACC CCGAG GA-3′, R: 5′- GCTTC TCGTT GGTCA GACCG AT-3′); *NT-3* (F: 5′- CTACT ACGGC AACAG AGACG CT-3′, R: 5′- GGTGA GGTTC TATTG GCTAC CAC-3′); *Gapdh* (F: 5′-AAGGT CATCCCAGAG CTGAA-3′, R: 5′- CTGCT TCACC ACCTT CTTGA-3′).

## Western blot and coimmunoprecipitation assay

Cells or muscles dissected from limb were lysed in the cell lysis buffer containing 50 mM Tris-HCl, pH 7.4, 150 mM NaCl, 2 mM EDTA, 1 mM PMSF, 50 mM NaF, 1% NP-40, 2% SDS, 0.5% sodium deoxycholate, 20% glycerol, 0.1% sodium vanadate and 1% protease inhibitor cocktail (Sigma-Aldrich Cat# 046931590001)[24]. Lysates were centrifuged at 12000 g at 4 °C for 20 min, and supernatants were diluted with 4× loading buffer (20% Tris-HCl, pH 8.8, 8% SDS, 8% β-mercaptoethanol, 0.04% bromophenol blue, 40% glycerol) and boiled for 10 min. Equal amounts of protein were separated by SDS-PAGE and transferred to nitrocellulose membranes (Bio-Rad Cat# 1620112). The membrane was blocked with 5% nonfat milk for 1 h at room temperature and then incubated with primary antibody at 4 °C overnight. After washing with 0.1% Tween in TBS three times (10 min/each), the membrane was incubated with HRP-conjugated secondary antibodies for 1 h at room temperature. Membrane was washed and imaged and quantified by LI-COR Odyssey Infrared Imaging System (LI-COR biosciences, Model 2800). Band density was normalized to loading control.

For co-IP assay, cells were lysed by adding nine volumes of modified RIPA buffer (150 mM NaCl, 2.5 mM EDTA, 50 mM Tris-HCl, 50 mM NaF, 0.1% sodium vanadate, 1% PMSF and 1% protease inhibitor cocktail) and incubated with 15 μl anti-Flag beads (Sigma-Aldrich Cat# M8823) for 4–5 h or overnight at 4 °C. Proteins pulled down by beads were subjected to Western blot.

## Muscle strength and weight

The strength of limb muscles was measured by SR-1 hanging scale (American Weigh Scales)[102]. Briefly, the forelimbs were allowed to grip a square metal grid which was connected to a hanging scale. The hind limbs were suspended, and the tails were gently horizontally pulled until mice released the grip. Mice weight was recorded by a Mettler Toledo table-top portable weighing scale.

## Rotarod test

Mice were placed on a rod accelerated at an increasing speed from 4 to 40 rpm during 5 min. Testing mice were housed at the testing room for 2 h and then allowed to complete one training trial. Each mouse performed 3 times with 10 min intervals between trials for three consecutive days. The time of mice fall from the rod was recorded[103].

## Balance beam walking task

The 80 cm beam (24, 12 or 6 mm in width) was set 50 cm above the floor, and a dark box was placed at the distal end of the beam and filled with bedding material to attract the mice to run across the beam. Mice were habituated in the testing room 2 h before performing the task. During training, a mouse was placed at the start point, and a light at the start point was turned on to encourage the mouse to cross the beam to the dark box. Each training contains three trails with 15 min intervals when the beam was changed to a narrower size, and the training was repeated at the same time next day. The beam and the box were cleaned with 70% alcohol and paper tissue before the next trail. Test was carried out on the third day with video recording. The time to across the beam and number of side slips were quantified[50].

## CTB injection and staining

2.5 μl of 0.25 mg/ml cholera toxin subunit B was injected into TA/EDL muscles. Five days later, mice were perfused and L3 DRGs were isolated, which were proceeded to stain with anti-CTB and/or anti-PV antibody[46]. Images were taken by Zeiss LSM 810 motorized confocal microscope. The number of CTB-positive neurons and soma size were quantified by ImageJ. Cells with soma size larger than 1000 μm² were quantified[46].

## Plasmid construction and recombinant protein production

cDNA encoding the APP E1 or E2 domain was amplified from full length *App* whereas that for LDLa was amplified from full length *Lrp4*. They were subcloned into p3 × Flag-*myc*-CMV™-24 between Hind III and EcoR I, and recombinant proteins were produced and purified[88,104]. HEK293T cells (ATCC Cat# CRL-3216) were cultured in DMEM supplemented with 10% FBS and 1 × penicillin/streptomycin and transfected with respective constructs with 0.1% polyethyleneimine (Polysciences

Cat# 24765, molecular weight 40,000) as reported[105]. 12 h after transfection, culture medium was changed to serum free medium and cultured for additional 36 h. The medium was collected and concentrated with Amicon centrifugal filter unit (Millipore Cat# UFC801008, UFC80300). One milliliter concentrated medium then incubated with 40 µl anti-Flag M2 agarose beads (Sigma-Aldrich Cat# A2220) at 4 °C overnight. Beads were washed with pre-cold TBS buffer (50 mM Tris HCL, 150 mM NaCl, pH 7.4) for three times. Then beads were incubated with 50 µl 0.1 M Glycine HCl (pH 3.0) for 5 min in the shaker at room temperature. 1000 g centrifuged for 1 min, and transferred the supernatant into a new tube added with 10 µl neutralized buffer (0.5 M Tris, 1.5 M NaCl, pH 8.0). Beads were washed with 50 µl 0.1 M Glycine HCl for two times. The recombinant proteins were eluted with 200 µl, 150 ng/µl 3 × Flag peptide (Sigma-Aldrich Cat# F4799), and concentrated with 10 kDa Amicon Ultra centrifugal filter (Millipore Cat# UFC901008) and kept at −80 °C at a concentration of 50 ng/µl in 50% glycerol solution till usage. The recombinant proteins APP E1, E2 and LRP4 LDLa were injected into EDL and TA muscles, 2.5 µg per mouse[106]. One week later, TA and EDL muscles were collected for histological analysis.

### AAV virus and DRG injection

*App* or *Aplp2* shRNAs were cloned into AAV-U6-sgRNA-hSyn-mCherry. The sequences were as follows: sh*App*-1, 5′-GCACA TGAAT GTGCA GAATG G-3′, sh*App*-2, 5′-GCACT AACTT GCACG ACTAT G-3′; sh*Aplp2*-1, 5′-CGATT ACAAT GAGGA GAATC CAACC GAAC-3′, sh*Aplp2*-2, 5′-ATGAA GGCTC TGGAA TGGCA GAACA AGAC-3′[407,108]. Retroviruses were produced with pAAV vector, pAAV-RC2 and pHelper (ratio of 2:1:1) co-transfected into cultured HEK293T cells by polyethyleneimine[109]. The medium was collected at 48, 72 and 96 h after transfection, filtered by 0.22 µm filters, and centrifuged at 20,000 × g for 2 h at 4 °C (Beckman, SW27 rotor). The virus-containing pellets were resuspended in 100 µl PBS, and injected into DRGs[110]. Adult mice were anaesthetized; L3-L5 DRGs were exposed by stereotactic surgery and injected with 3 µl virus ($1.5 \times 10^{11}$ vg/ml) into epineurium or DRG capsule by a sharp glass micropipette attached to a Hamilton syringe at a rate of 0.5 µl/min. After wound suturing, mice were placed on a heated blanket until recovered. Four weeks after the injection, the DRG and muscle were isolated for RT-qPCR and histological analysis.

### Statistical analysis

Data were analyzed by unpaired two-tailed *t* test (two independent data comparisons) and one-way ANOVA (multiple comparisons). Sample size was determined based on literature and previous experience[31,65,111]. Unless otherwise indicated, data were mean ± SEM and described in figure legends. Statistical significance was determined by GraphPad Prism 6.0 and labeled as: *$p < 0.05$; **$p < 0.01$; ***$p < 0.001$.

### Reporting summary

Further information on research design is available in the Nature Portfolio Reporting Summary linked to this article.

## Data availability

All data supporting the findings of this study are available in the article and its supplementary information. Source data are provided with this paper as Source Data file. Source data are provided with this paper.

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

## Acknowledgements

We thank members of Mei/Xiong Lab for critical comments. This study was supported by funds from Case Western Reserve University to LM and WCX.

## Author contributions

R.C., Conceptualization, Resources, Data curation, Software, Formal analysis, Validation, Methodology, Writing - original draft and editing; P.C., H.W., Resources, Data curation, Formal analysis, Validation, Methodology; H.J., Data curation, Investigation, Methodology; H.Z., J.P., Data curation, Investigation, Software; G.X., Data curation, Formal analysis, Methodology; B.L., Z.Y., Data curation, Methodology; W.X., Conceptualization, Project administration; L.M., Conceptualization, Resources, Supervision, Funding acquisition, Writing, Project administration. All authors reviewed and approved the final draft of the manuscript.

## Competing interests

The authors declare no competing interests.
