## [Peer Review File · Nature Communications]

Intrafusal-fiber LRP4 for muscle spindle formation and maintenance in adult and aged animalsREVIEWER COMMENTS

Reviewer #1 (Remarks to the Author):

Cao et al. report a role of Lrp4 derived from intrafusal fibers in the formation and maintenance of the sensory innervation of muscle spindles. The work is novel and in general well executed, and it will be of interest for a broad readership. It does not only address spindle development, but also spindle maintenance in aging. I have a number of points that need to be addressed by additional experiments before publications. Particularly important is point 6, which I consider it absolutely essential to address.

In addition, I find relatively many sentences that are awkward, lack words or contain unnecessarily repeated words. The manuscript could be improved by detailed copy editing. I list some of these awkward sentences at the end of my review.

1) Fig. 1: Lrp4 is highly expressed in intrafusal fibers. However, apparently LRP4-CreERT does not recombine all fibers of the spindle. The spindle contains several types of intrafusal fiber types (bag, chain): do all of these fiber types expressing Lrp4 at high levels?

2) A recent snRNAseq paper has identified a sensory compartment in the spindle, i.e. a particular set of genes that are expressed at higher levels in nuclei close to the annulospiral endings (Kim et al., Nat Commun. 2020 Dec 11;11(1):6375. doi: 10.1038/s41467-020-20064-9.). Are Lrp4 transcripts preferentially found in this sensory compartment?

3) Line 220 Annulospiral endings per spindle were decreased from 10.438 ...(what is the unit of 10.438)?

4) Fig. 5: A quantification of PV+ neuron numbers in DRGs of controls and mutant mice should be included.

5) Fig. 5: the authors should mention in the description of Fig. 5 the appearance/time course of deficits in the NMJ after Lrp4 deletion in the muscle. Can NMJ deficits cause or contribute to the behavioral changes described?

6) The picture shown in Fig. 6b indicates that a relatively small proportion of DRG neurons were infected. Nevertheless, the phenotype on the annulospiral endings in the spindle (Fig. 6e) are observed in half of the spindles. In order to address this properly, the number of PV+ neurons expressing mCherry has to be quantified in the same animals in which the % of spindles with annulospiral endings are determined.

7) Lines 344-353: As LRP4 was required for the maintenance of muscle spindle, we wondered whether...

I do not understand the experiment described: why is Lrp4 stained with 555-LRP4 and not with an anti-Lrp4 antibody? Why does 555-LRP4 stain endogenous LRP4?

8) Please introduce the abbreviations used, e.g. ECD.

Text:

1) The abstract and some of the text is confusing, because the authors do not distinguish between the muscle spindle (i.e. the structure containing several types of intrafusal fibers surrounded by a capsule; the fibers are innervated by afferent proprioceptive and efferent gamma motor neurons) and the endings of the afferent proprioceptive neurons.

e.g. line 151 These results suggest that Lrp4 mutation prevents spindle development without altering intrafusal fiber differentiation.

I assume that the authors mean to say: Lrp4 mutation prevents development of the annulospiral or

sensory endings and spindle function.

2) Line 141: These results demonstrate that spindles are impaired spindles in the absence of LRP4,

3) Line 225: please replace time of experiment with duration of experiment

4) Line 278: such as APLP2 that resembles the spatial expression pattern of APP in the peripheral nerve system (change to: APLP2 that displays a similar spatial expression in the peripheral nervous system)

5) Line 356: EDL muscles of 3 M, 12 M and 24 M were .. Please rephrase: 24 M-old mice

6) Line 387: In *Lrp4* mutant mice, sensory nerve terminals were disorganized, discontinuous or fragmented in *Lrp4* null mutant mice (please remove part of the sentence written in italics).

Reviewer #2 (Remarks to the Author):

This is a nice informative manuscript that reveals a novel role for the LDL receptor family member LRP4 in the development and maintenance of proprioceptive muscle spindles. The study has been methodically conducted, the data are strong and the conclusions solid. This paper will be of broad interest and contribute major new insights into the mechanisms of muscle innervation.

My only criticism concerns the extent of the discussion and the limitations of the study.

I feel it is important to stress that the role of LRP4 in mice appears to be strikingly different from its function in humans and bovines. Specifically, LOF mutations in LRP4 in mice are perinatally lethal, as the mice lack neuromuscular junctions and consequently die from failure to breathe. By contrast, LOF mutations in LRP4 cause Cenani-Lenz syndrome in humans (Li et al., *Am J Hum Genet.* 2010 May 14; 86(5): 696–706) and mulefoot disease in mice (Johnson et al., *Genomics* 2006 Nov;88(5):600-9). These findings, and the possible reasons for the discrepancy, should be discussed in depth. In addition, the following reference should be added to the discussion on the role of LRP4 in Wnt signaling: Johnson, E., et al., *Hum Mol Genet.* 2005 Nov 15;14(22):3523-38

Reviewer #1 (Remarks to the Author):

Cao et al. report a role of Lrp4 derived from intrafusal fibers in the formation and maintenance of the sensory innervation of muscle spindles. The work is novel and in general well executed, and it will be of interest for a broad readership. It does not only address spindle development, but also spindle maintenance in aging. I have a number of points that need to be addressed by additional experiments before publications. Particularly important is point 6, which I consider it absolutely essential to address.

Response – We thank the reviewer for her or his comments that “The work is novel and in general well executed, and it will be of interest for a broad readership...”. As described below, we have performed additional experiments to address the concern of point 6.

In addition, I find relatively many sentences that are awkward, lack words or contain unnecessarily repeated words. The manuscript could be improved by detailed copy editing. I list some of these awkward sentences at the end of my review.

Response – All identified sentences have been revised. In addition, changes have been made in the entire manuscript for better description of results, discussion points and to eliminate grammatic and typographic errors.

1) Fig. 1: Lrp4 is highly expressed in intrafusal fibers. However, apparently LRP4-CreERT does not recombine all fibers of the spindle. The spindle contains several types of intrafusal fiber types (bag, chain): do all of these fiber types expressing Lrp4 at high levels?

Response – Good question. In additional experiments, we labeled intrafusal fibers with the bag fiber-specific monoclonal antibody S46¹. tdTomato fluorescence intensity appeared to be variable in S46-positive fibers (Fig. 1k). We have attempted to label chain fibers with the antibody A4.74², but failed despite repeated attempts. Nevertheless, tdTomato was present in intrafusal fibers that were negative for S46; they were presumably chain intrafusal fibers. The tdTomato fluorescence was also variable in these fibers (Fig. 1k). These results suggest that LRP4 expression was uneven in different intrafusal fibers. These results have been described in the revised manuscript (page 6).

2) A recent snRNAseq paper has identified a sensory compartment in the spindle, i.e. a particular set of genes that are expressed at higher levels in nuclei close to the annulospiral endings (Kim et al., Nat Commun. 2020 Dec

11;11(1):6375. doi: 10.1038/s41467-020-20064-9.). *Are Lrp4 transcripts preferentially found in this sensory compartment?*

Response – Thank you for alerting us on this paper. *Lrp4* was not present in the set of genes in the nuclei close to the annulospiral endings in single nucleus-seq analysis by Kim et al. (2020). *Lrp4* was present once only in the set of genes in the nuclei close to the spindle NMJ (formed by γ -motor neurons). However, interpretation of these results should be careful because the latter set of genes lacks other NMJ markers such as *MuSK*, *DOK7* or *rapsyn*. This was likely due to the low detection efficiency in the Kim et al. study that detected only 1000-2000 mRNAs per nucleus. Future, more efficient studies are warranted to determine detailed gene profiles of nuclei close to the annulospiral endings. We discussed these points in the revised manuscript (page 19).

3) *Line 220 Annulospiral endings per spindle were decreased from 10.438 ... (what is the unit of 10.438)?*

Response – We apologized for being unclear. The unit is the number of the spiral endings per spindle (with one circle scored as 1). This has been described in the revised manuscript (page 10, page 14, page 15).

4) *Fig. 5: A quantification of PV+ neuron numbers in DRGs of controls and mutant mice should be included.*

Response – As suggested, PV⁺ neuron numbers in DRGs have been quantified and shown in revised Fig. 5f and text (page 11). There was no difference among the three groups.

5) *Fig. 5: the authors should mention in the description of Fig. 5 the appearance/time course of deficits in the NMJ after Lrp4 deletion in the muscle. Can NMJ deficits cause or contribute to the behavioral changes described?*

Response – As suggested, results of Fig. 5 are now described in more details (page 12). Can NMJ deficits cause or contribute to the behavioral changes described? This is a great question. Assessment of proprioceptive behavior requires muscle strength. Impaired proprioception has been documented as a secondary effect in disorders that impinge on α -MN NMJs such as myasthenia gravis^{3,4}. Nevertheless, proprioceptive behaviors were reported normal in muscle dystrophy mouse models with mild muscle weakness and bodyweight loss⁵. Evidently, LRP4 is critical to the formation and maintenance of α -MN NMJs^{6,7}. *Lrp4* null mutation impaired development of sensory terminals (Fig. 2, 3); because of the neonatal lethality, it prevents behavioral studies of

proprioceptive function. However, proprioceptive behaviors were compromised in *Lrp4* conditional mutant mice (Fig. 5), associated with severe deficits in spindle morphology and sensory synapses (Fig. 4). A parsimonious interpretation of these results supports a role of LRP4 in spindle formation and maintenance. These points are discussed in the revised manuscript (page 19-20).

6) *The picture shown in Fig. 6b indicates that a relatively small proportion of DRG neurons were infected. Nevertheless, the phenotype on the annulospiral endings in the spindle (Fig. 6e) are observed in half of the spindles. In order to address this properly, the number of PV+ neurons expressing mCherry has to be quantified in the same animals in which the % of spindles with annulospiral endings are determined.*

Response – Good suggestion. As suggested, we quantified DRG PV+ neurons that were positive for mCherry (that was expressed by the AAV virus). As shown in revised Fig. 6d and e, more than 80% of DRG PV+ neurons were also positive for mCherry, indicating a high infection efficiency.

7) *Lines 344-353: As LRP4 was required for the maintenance of muscle spindle, we wondered whether...*

I do not understand the experiment described: why is Lrp4 stained with 555-LRP4 and not with an anti-Lrp4 antibody? Why does 555-LRP4 stain endogenous LRP4?

Response – We are sorry for the confusion. The reviewer was correct that “555-LRP4” should read “555-anti-LRP4”. Related text has been revised to correct this error (page 16, 45).

8) *Please introduce the abbreviations used, e.g. ECD.*

Response – ECD has now been defined, as suggested.

Text:

1) *The abstract and some of the text is confusing, because the authors do not distinguish between the muscle spindle (i.e. the structure containing several types of intrafusal fibers surrounded by a capsule; the fibers are innervated by afferent proprioceptive and efferent gamma motor neurons) and the endings of the afferent proprioceptive neurons.*

Response – Good suggestion. Changes have been made to address this concern as indicated below and also in abstract and discussion.

e.g. line 151 These results suggest that Lrp4 mutation prevents spindle

development without altering intrafusal fiber differentiation.

I assume that the authors mean to say: Lrp4 mutation prevents development of the annulospiral or sensory endings and spindle function.

Response – The sentence has been changed to “LRP4 mutation impairs development of sensory endings”, as suggested. Sensory endings in embryonic stage have not yet become “annulospiral” and “spindle function” was not measured until Fig. 4. These two terms were omitted in the sentence.

2) *Line 141: These results demonstrate that spindles are impaired spindles in the absence of LRP4,*

Response – The sentence has been changed to “These results demonstrate that the sensory endings are impaired in the absence of LRP4”, as suggested.

3) *Line 225: please replace time of experiment with duration of experiment*

Response – Replaced as suggested.

4) *Line 278: such as APLP2 that resembles the spatial expression pattern of APP in the peripheral nerve system (change to: APLP2 that displays a similar spatial expression in the peripheral nervous system)*

Response – This sentence has been revised.

5) *Line 356: EDL muscles of 3 M, 12 M and 24 M were .. Please rephrase: 24 M-old mice*

Response – Revised it as suggested.

6) *Line 387: In Lrp4 mutant mice, sensory nerve terminals were disorganized, discontinuous or fragmented in Lrp4 null mutant mice (please remove part of the sentence written in italics).*

Response – Removed as suggested.

Reviewer #2 (Remarks to the Author):

This is a nice informative manuscript that reveals a novel role for the LDL receptor family member LRP4 in the development and maintenance of proprioceptive muscle spindles. The study has been methodically conducted, the data are strong and the conclusions solid. This paper will be of broad

interest and contribute major new insights into the mechanisms of muscle innervation.

Response – We thank the reviewer for the comments of “This is a nice informative manuscript that reveals a novel role...” and “This paper will be of broad interest and contribute major new insights into the mechanisms of muscle innervation”.

My only criticism concerns the extent of the discussion and the limitations of the study.

I feel it is important to stress that the role of LRP4 in mice appears to be strikingly different from its function in humans and bovines. Specifically, LOF mutations in LRP4 in mice are perinatally lethal, as the mice lack neuromuscular junctions and consequently die from failure to breathe. By contrast, LOF mutations in LRP4 cause Cenani-Lenz syndrome in humans (Li et al., Am J Hum Genet. 2010 May 14; 86(5): 696–706) and mulefoot disease in mice (Johnson et al., Genomics 2006 Nov;88(5):600-9). These findings, and the possible reasons for the discrepancy, should be discussed in depth. In addition, the following reference should be added to the discussion on the role of LRP4 in Wnt signaling: Johnson, E., et al., Hum Mol Genet. 2005 Nov 15;14(22):3523-38

Response – These are insightful comments and suggestions. As the reviewer pointed out, yes, earlier studies suggest that LOF mutations in LRP4 causes Cenani-Lenz syndrome in human and mulefoot disease in bovine, unlike mice where LRP4 null mice die neonatally because of the lack of the NMJ. LRP4 is a transmembrane protein with a large extracellular domain (ECD) and a small intracellular domain (ICD) that is expressed in many tissues including muscles, brain, kidney and bone⁸. We showed previously that the soluble ECD is able to function as a receptor (albeit with lower efficacy than the full length LRP4) for agrin to activate MuSK⁹. In accord, mutant mice or bovine expressing the ECD without the transmembrane domain or ICD are able to survive but display syndactyly phenotypes^{10,11}, suggesting the ICD may be necessary for signaling to prevent mulefoot or syndactyly. Most LRP4 mutations in patients with Cenani-Lenz syndrome (CLS) are recessive and believed to alter its expression or function¹²⁻¹⁵. In mice, neonatal lethality can be caused by null mutation or mutations missing a critical region in the extracellular domain (ECD) such as *mitt* and *mte*⁶. Several mutations have been identified in human that produce truncated LRP4 mutant proteins that lack a segment of the ECD (c.2401 A>T, c.3062 del C, and c.199-200ins GATTCAG)^{12,14} or the transmembrane domain (c.4952-4987 del)¹² (Table 1). Although none of them were homozygous, however compound heterozygous alterations (c.2401 A>T and c.3062 del C or c.4952-4987 del and c.199-200ins GATTCAG) caused prenatal lethality^{12,14}. A single amino acid

missense mutation in the first $\beta 1$ propeller domain (c.1585G>A) causes neonatal lethality¹² (Table 1). These results have the following implications with an assumption that the mutant proteins are expressed in a manner dependent on gene-dosage. First, compound, severe loss-of-function LRP4 mutations cause pre- or neonatal lethality, in agreement with mouse studies. Second, LRP4 ECD is critical for its function. Third, the lethality of the compound heterozygous mutations (c.4952-4987 del and c.199-200ins GATTCAG) suggest that a single copy of the ECD is not sufficient for signal transduction. Alternatively, the ECD truncation mutant protein produced by c.199-200ins GATTCAG could be a dominant negative. However, homozygous mutation of c.289G>T that leads to a premature stop codon at amino acid 97 (p.E97X) at the very beginning of the large extracellular domain was not lethal¹⁶, which remains to be a puzzle difficult to explain.

These points have now been discussed in the revised manuscript (page 18, 19). As suggested, the Johnson et al. 2005 paper and two additional papers (Li et al., 2010; Johnson et al., 2006) have been cited. In addition, a review on LRP4 by Herz et al. (2002) in the introduction.

Table 1. LRP4 mutations in human and mouse

Human	Mutation	Nucleotide	Protein	Genotype	Lethality	Reference
1. c.2401 A>T	p.K801X	801aa	Truncated in $\beta 1$ and after	hetero (1+2)	Prenatal lethal	Lindy et al., 2014
2. c.3062 del C	p.S1020Qfs*27	1047aa	Truncated in $\beta 3$	No report on homo	N/A	Lindy et al., 2014
3. c.4952-4987del	p.V1651Dfs*40	1691aa	Truncated after $\beta 4$	hetero (3+4)	Prenatal lethal	Li et al., 2010
4. c.199_200ins GATTCAG	p.I67Rfs*10	77aa	Truncated in LDLa	No report on homo	N/A	Li et al., 2010
5. c.289G>T	p.E97X	97aa	Truncated in LDLa	homo	Severe CLS	Kariminejad et al., 2013
6. c.1585G>A	p.D529N	1905aa (D>N)	Missense in $\beta 1$	homo	Die at first day or prenatal	Li et al., 2010
Mouse						
1. Lrp4 mitt	Q377ter Splice site mutate	377aa	Truncated after LDLRa	homo	Neonatal lethal	Weatherbee et al., 2006
2. Lrp4 mte	D1436G	1905aa (D>G)	Missense in $\beta 4$	homo	Neonatal lethal	Weatherbee et al., 2006
3. Lrp4 null	First exon is replaced with a neomycin stop cassette	N/A	null	homo	Perinatal lethal	Karner et al., 2010
4. LRP4 ECD	A stop codon into exon 36	1725aa	Truncated after ECD	homo	Not die	Johnson et al., 2005

References

- 1 Muller, K. A., Ryals, J. M., Feldman, E. L. & Wright, D. E. Abnormal muscle spindle innervation and large-fiber neuropathy in diabetic mice. *Diabetes* **57**, 1693-1701, doi:10.2337/db08-0022 (2008).
- 2 Radovanovic, D., Peikert, K., Lindstrom, M. & Domellof, F. P. Sympathetic innervation of human muscle spindles. *J Anat* **226**, 542-548, doi:10.1111/joa.12309 (2015).
- 3 Cazzato, G. & Walton, J. N. The pathology of the muscle spindle. A study of biopsy material in various muscular and neuromuscular diseases. *J Neurol Sci* **7**, 15-70, doi:10.1016/0022-510x(68)90003-8 (1968).
- 4 Swash, M. & Fox, K. P. The pathology of the muscle spindle in myasthenia gravis. *J Neurol Sci* **26**, 39-47, doi:10.1016/0022-510x(75)90112-4 (1975).
- 5 Wang, J. *et al.* Cell-autonomous requirement of TDP-43, an ALS/FTD signature protein, for oligodendrocyte survival and myelination. *Proc Natl Acad Sci U S A* **115**, E10941-E10950, doi:10.1073/pnas.1809821115 (2018).
- 6 Weatherbee, S. D., Anderson, K. V. & Niswander, L. A. LDL-receptor-related protein 4 is crucial for formation of the neuromuscular junction. *Development* **133**, 4993-5000, doi:10.1242/dev.02696 (2006).
- 7 Barik, A. *et al.* LRP4 is critical for neuromuscular junction maintenance. *J Neurosci* **34**, 13892-13905, doi:10.1523/JNEUROSCI.1733-14.2014 (2014).
- 8 Herz, J. & Bock, H. H. Lipoprotein receptors in the nervous system. *Annu Rev Biochem* **71**, 405-434, doi:10.1146/annurev.biochem.71.110601.135342 (2002).
- 9 Wu, H. *et al.* Distinct roles of muscle and motoneuron LRP4 in neuromuscular junction formation. *Neuron* **75**, 94-107, doi:10.1016/j.neuron.2012.04.033 (2012).
- 10 Johnson, E. B., Hammer, R. E. & Herz, J. Abnormal development of the apical ectodermal ridge and polysyndactyly in *Megf7*-deficient mice. *Hum Mol Genet* **14**, 3523-3538, doi:10.1093/hmg/ddi381 (2005).
- 11 Johnson, E. B., Steffen, D. J., Lynch, K. W. & Herz, J. Defective splicing of *Megf7/Lrp4*, a regulator of distal limb development, in autosomal recessive mulefoot disease. *Genomics* **88**, 600-609, doi:10.1016/j.ygeno.2006.08.005 (2006).
- 12 Li, Y. *et al.* LRP4 mutations alter Wnt/beta-catenin signaling and cause limb and kidney malformations in Cenani-Lenz syndrome. *Am J Hum Genet* **86**, 696-706, doi:10.1016/j.ajhg.2010.03.004 (2010).
- 13 Khan, H. *et al.* Novel variants in the LRP4 underlying Cenani-Lenz Syndactyly syndrome. *J Hum Genet* **67**, 253-259, doi:10.1038/s10038-021-00995-x (2022).
- 14 Lindy, A. S. *et al.* Truncating mutations in LRP4 lead to a prenatal lethal form of Cenani-Lenz syndrome. *Am J Med Genet A* **164A**, 2391-2397, doi:10.1002/ajmg.a.36647 (2014).
- 15 Sukenik Halevy, R. *et al.* Mutations in the fourth beta-propeller domain of LRP4 are associated with isolated syndactyly with fusion of the third and fourth fingers. *Hum Mutat* **39**, 811-815, doi:10.1002/humu.23417 (2018).
- 16 Kariminejad, A. *et al.* Severe Cenani-Lenz syndrome caused by loss of LRP4 function. *Am J Med Genet A* **161A**, 1475-1479, doi:10.1002/ajmg.a.35920 (2013).

REVIEWERS' COMMENTS

Reviewer #1 (Remarks to the Author):

My concerns were appropriately addressed in the revised manuscript.

Reviewer #2 (Remarks to the Author):

All the revisions are satisfactory and have substantially clarified and improved the manuscript. I have no further concerns.